# The probabilistic hydrological MARCS[HYDRO] (the MARkov Chain System) model: Its structure and core version 0.2

Elena Shevnina[1], Andrey Silaev[2]

[1]Finnish Meteorological Institute, Helsinki, FI-00560, Finland
[2]National Research University Higher School of Economics, Nizhny Novgorod, 603155, Russia.

*Correspondence to: Elena Shevnina (elena.shevnina@fmi.fi)*

**Abstract.** The question of the environmental risks of social and economic infrastructure has recently become apparent due to an increase in the number of extreme weather events. Extreme runoff events include floods and droughts. In water engineering, extreme runoff is described in terms of probability and uses methods of frequency analysis to evaluate an exceedance probability curve (EPC) for runoff. It is assumed that historical observations of runoff are representative for the future; however, trends in the observed time series doubt this assumption. The paper describes a probabilistic hydrological MARCS[HYDRO] model that can be applied to predict future runoff extremes. The MARCS[HYDRO] model simulates statistical estimators of a multi-year runoff in order to perform future projections in a probabilistic form. Projected statistics of the meteorological variables available in climate scenarios force the model. This study introduces the new model's core version and provides a user guide together with an example of the model set-up in a single case study. In this case study, the model simulates the projected EPCs of annual runoff under three climate scenarios. The scope of applicability and limitations of the model's core version 0.2 are discussed.

## 1 Introduction

Streamflow runoff serves as a water resource for humans, food production and energy generation while the risks of water-sensitive economics are usually connected to runoff extremes. In fact, the runoff extremes are always connected to a human activity since they do not exist in a natural water cycle. Engineering science considers the runoff extremes as critical values of runoff that lead to the damage of infrastructure or water shortages, and it introduces the extremes in terms of probability. In particular, in water engineering the runoff extremes are evaluated from the tails of exceedance probability curves (EPCs) that are used in risk assessment for water infrastructure and decision-making in cost-loss situations (Mylne, 2002; Murphy, 1977, 1976). The EPC of multi-year runoff allows the estimation of the runoff extremes and supports the designing of building constructions, bridges, dams, withdrawal systems etc.

Modern hydrology uses two approaches to evaluate the runoff extreme with their exceedance probability: conceptual modelling (Lamb, 2006) and a frequency analysis (Kite, 1977; Benson, 1968; Kritsky and Menkel, 1946). In the conceptual modelling approach, synthetic runoff series are simulated from meteorological series in order to calculate the runoff values

of a chosen exceedance probability (Arheimer and Lindström, 2015; Veijalainen et al., 2012; Seibert, 1999). In the frequency-analysis approach, historical yearly time series of runoff are used to evaluate statistical estimators, that is, mean value, the coefficient of variation (CV) and the coefficient of skewness (CS) (van Gelder, 2006). These estimators are applied to calculate the runoff values with the exceedance probability (Guidelines SP 33-101-2003, 2004; Guidelines, 1984; Bulletin 17–B, 1982) needed to support the designing of roads, dams, bridges or water-withdrawal stations. The basic assumption of this approach is that the future risks during an infrastructure's operational period are equal to the risks estimated from the past observations. The runoff extremes are simply extrapolated for the next 20–30 years on the assumption that the past observations are representative of the future: the "stationarity" assumption (Madsen et al., 2013).

The number of weather extremes – including hurricanes, wind, rain and snow storms, floods and droughts – has increased (Vihma, 2014; Wang and Zhou, 2005, Manton et al., 2001). Historical time series of many climate variables show evident trends, which are statistically significant, and the series of streamflow runoff are among others (Wagner et al., 2011; Dai et al., 2009; Milly et al., 2005). Rosmann et al. (2016) applied the Mann–Kendall test to analyse a time series of daily, monthly and yearly river discharges for the last four decades. The highest number of trends was detected for the yearly time series of annual runoff. The statistically significant trends are founded on historical time series, thus the water engineers and managers are motivated to revise the basic stationarity assumption that lies behind infrastructures' risk assessment since the past observations are representative of the future (Madsen et al., 2013; Kovalenko, 2009; Milly at al., 2008).

In this paper, we described a method that combines the conceptual modelling and frequency analysis in order to estimate the runoff extremes in a changing climate. The method adapts the theory of stochastic systems to the water engineering practice and it was further named as advanced of frequency analysis (AFA). It was introduced by Kovalenko (1993) and relied on the theory of stochastic systems (Pugachev et al., 1974). The basic idea behind the method is to simulate the statistical estimators of multi-year runoff (annual, minimal and maximal runoff) from the statistical estimators of precipitation and air temperature on a climate scale (Budyko and Izrael, 1991). The simulated statistical estimators of runoff are used to construct EPCs with distributions from the Pearson system (Pearson, 1895). Kovalenko (1993) suggested modelling the EPCs within Pearson Type III distribution based on a traditional practice in water engineering (Rogdestvenskiy and Chebotarev, 1974; Matalas and Wallis, 1973; Sokolovskiy, 1964). However, the distribution can be also chosen by fitting (Laio et al., 2009), defined in accordance with local hydrological guidelines (Bulletin 17-B, 1982) or somehow more advanced (Andreev et al., 2005).

A linear "black box" (or a "linear filter model" ) with stochastic components is suggested as a catchment-scale hydrological model (Kovalenko, 1993). For this linear model, the theory of stochastic systems provides methods to direct the simulation of probability distributions for a random process (Pugachev et al., 1974). The theory of stochastic systems is applied to analyse and predict runoff extremes on various time scales, ranging from days (Rosmann and Domínguez, 2017) to a season (Dominguez and Rivera, 2010; Shevnina, 2001) and climate scales (Shevnina et al., 2017; Kovalenko, 2014, Viktorova and Gromova, 2010). The AFA approach is a simplification of the theory of stochastic systems on a climate scale. Kovalenko et al. (2010) gave guidelines for water engineers to estimate the runoff extremes in a changing climate.

AFA was suggested about 30 years ago; however, a full description of this approach has still not been published in English. Moreover, the previous publications in Russian contain many typewriting mistakes in the formulas (Kovalenko, 1993; Kovalenko et al., 2006), and this makes understanding them troublesome, even for native Russians. In this paper, the theory and assumptions of the AFA approach were formulated "step-by-step" (see Annex 1), and the formulas behind the core of the probabilistic hydrological model MARCS$^{HYDRO}$ were accepted for the new version, version 0.2 (see Section 1). This model core allows the

prediction of a skewness parameter of Pearson Type III distribution. An example of the model set-up, forcing and output for a case study of the Iijoki river is given in Section 2. The main features of the model and the limitations of the AFA method are formulated in the Discussions section in order to better place the MARCS$^{HYDRO}$ model among other hydrological models.

## 2 Model description

The probabilistic hydrological MARCS$^{HYDRO}$ model consists of six blocks (Shevnina, 2015). Fig. 1 shows the tools for data

analysis grouped into blocks: two blocks for the analysis and screening of observed data (DPB and DSB); a block with the model parametrization, cross-validation and hind casts (PHP); a block to visualise the model's results (VAB); and a block with socio-economic applications (EAB). Shevnina and Gaidukova (2017) provided details about the algorithms already implemented in each block in the model. In this paper, only version 0.2, for the model's core, is introduced. The formulas behind the model's core version 0.1 is published in the annex to the work of Shevnina et al. (2017).

The MARCS$^{HYDRO}$ model simulates three non-central statistical moments of multi-year runoff based on the means of precipitation calculated over a period of 20–30 years. Now, the model's application is only limited by a prediction on the climate scale. The development of a socio-economic infrastructure also needs the climate-scale prediction of river runoff (Milly et al., 2008) because water extremes, such as floods and droughts, lead to economical losses. The AFA approach has found practical applications in building constructions (Shevnina et al., 2017; Kovalenko, 2009). The MARCS$^{HYDRO}$ model

allows the "quick analysis" of the runoff extremes under different climate scenarios. The model needs less computational resources because it simulates the parameters of the distribution while the conceptual hydrological models simulate the runoff time series.

The MARCS$^{HYDRO}$ model parametrization, cross-validation and hindcasts need observations of the river water discharges of a hydrological network for a period in the past (Kovalenko, 1993). For the cross-validation, the yearly time series of river

runoff are split into two sub-periods, namely the training period and the control period (Shevnina, 2017). The splitting year corresponds to the year when the statistically significant difference in observations within two periods is detected by the Student and Kolmogorov-Smirnov tests (Kovalenko, 1993; Kovalenko et al., 2006). The description of the analysis and screening of the observed river runoff time series, as well as the model cross-validation procedure, fell outside the topics of this paper. We focused on the equations behind the model's core version 0.2 and its limitations.

## 2.1 Model input

Two blocks of the MARCS$^{HYDRO}$ model are needed to analyse and screen the observations. The time series of river runoff and precipitation are required for the period as longer as possible. However, the length of yearly time series on water discharges does not usually exceed 80–90 years. Hydrological yearbooks or runoff data sets provide observations at sites of national hydrological networks, and the river runoff is expressed as a volumetric flow rate (water discharge, m$^3$s$^{-1}$). In the data preparation block of the model, the volumetric flow rate (m$^3$s$^{-1}$) is converted to a specific water discharge (ARR, mm year$^{-1}$):

$$ARR = 1000 \ Q \ T / A,$$

where $Q$ is a yearly average water discharge (m$^3$s$^{-1}$), $T$ is the number of seconds in a year and $A$ is the catchment area (m$^2$). In the data screening block of the model, the yearly time series of ARR are used in the analysis of homogeneity and trends (Dalmeh and Hall, 1990) and to define a period for the model parametrization (called "a reference period" by Shevnina et al., (2017)). Then, the reference three non-central moments $m_k$ ( $m_k = 1/n \sum_{i=1}^{n} DR_i^k$ for $k$ = 1, 2, 3) are estimated from time series of ARR using the method of moments (van Gelder et al., 2006).

The observations on precipitation are collected from meteorological sites, and they may be interpolated into grids in order to better estimate a precipitation rate over a river basin area. In the data preparation block of the model, the mean annual precipitation rate (mm year$^{-1}$) is calculated from the observed yearly time series for the reference period. The mean annual precipitation rate for the future period can be calculated from an output of any global/regional climate model or even a set of models. In a study on the catchment scale, the time series of water discharges can be extracted from the Global Runoff Data Centre (GRDC) while the precipitation rate can be estimated from gridded data sets (Willmott and Robeson, 1995). These two data sets were used to perform an example of the model application on the Iijoki River basin.

## 2.2 Model cross-validation

The MARCS$^{HYDRO}$ model allows the simulation of the non-central moments of runoff that can be used for the construction of probability distribution (or an EPC), in other words, it provides a probabilistic form of prediction. The end product of the model is the probability density function (PDF) (or the EPC), and there are no simulated time series of runoff to compare with the observations. Kovalenko (1993) suggested comparing the simulated PDF with an empirical PDF by using known statistical tests such as the Kolmogorov-Smirnov test (Smirnov, 1948). In the PHB of the MARCS$^{HYDRO}$ model, a specific cross-validation procedure allows conclusions to be drawn about the model's validation and the quality of hindcasts. For the model's cross-validation, the observed time series of river runoff is divided into two sub-periods, namely the training period and the control period. The splitting year corresponds to the year when a statistically significant difference in mean values is estimated over two periods. In this study, we did not pay much attention to the cross-validation procedure since the model core version 0.2 is described in details in Shevnina et al. (2019).

## 2.3 The MARCS<sup>HYDRO</sup> model core

In our study, core version 0.2 of the probabilistic MARCS<sup>HYDRO</sup> model was suggested instead of version 0.1 (Shevnina et al., 2017). Version 0.2 allows the evaluation of the skewness parameter of the Pearson Type III distribution. In the new core, the non-central statistical moments of the ARR were calculated as follows:

$$m_1 = a - b_1 , \tag{1}$$

$$m_2 = -b_0 - 2m_1 b_1 + m_1 a , \tag{2}$$

$$m_3 = -2m_1 b_0 - 3m_2 b_1 + m_2 a , \tag{3}$$

where $m_1$, $m_2$ and $m_3$ are the moment estimates of the non-central statistical moments of the ARR; $a$, $b_0$, $b_1$ and $b_2$ are the parameters of the distributions of the Pearson equation (Andreev et al., 2005).

To set up the MARCS<sup>HYDRO</sup> model, observations of water discharges are needed. For the reference period (notated by low index $r$) the moments' estimates for the non-central moments ($m_{1r}$, $m_{2r}$, $m_{3r}$) were first calculated from observed times series of runoff (mm year$^{-1}$), then the non-central moments were used to evaluate the parameters of the Pearson equation $a$, $b_0$ and $b_1$:

$$a = 0.5 \left( 5m_{1r} m_{2r} - 4m_{1r}^3 - m_{3r} \right) / \left( m_{2r} - m_{1r}^2 \right) , \tag{4}$$

$$b_0 = 0.5 \left( m_{1r}^2 m_{2r} - 2m_{2r}^2 + m_{1r} m_{3r} \right) / \left( m_{2r} - m_{1r}^2 \right) , \tag{5}$$

$$b_1 = 0.5 \left( 3m_{1r} m_{2r} - 2m_{1r}^3 - m_{3r} \right) / \left( m_{2r} - m_{1r}^2 \right) . \tag{6}$$

Then, the parameters of the linear filter model (see Annex 1 for details) $\bar{c}, G_{\widetilde{N}}, G_{\widetilde{c}\widetilde{N}}$ denoted with a low index $r$, were calculated:

$$\bar{c}_r = \bar{N}_r / \left( a - b_1/2 \right) , \tag{7}$$

$$G_{\widetilde{N}r} = -2b_0 \bar{N}_r / \left( a - b_1/2 \right) , \tag{8}$$

$$G_{\widetilde{c}\widetilde{N}r} = b_1 \bar{N}_r / \left( a - b_1/2 \right) , \tag{9}$$

where $\bar{N}_r$ is the mean of annual precipitation rate (mm year$^{-1}$) estimated from observed time series as an average over any chosen reference period.

To force the MARCS<sup>HYDRO</sup> model, the outputs from global/regional-scale climate models are needed. Coupled Model Inter-comparison Project 5 (CMIP5; Taylor et al., 2012) is one collection of data sets that is available for climate-scale hydrological studies. Recently, the model only needs to be forced by a mean of precipitation (mm year$^{-1}$), evaluated for a future period of 20–30 years. A low index $pr$ indicated that the values were estimated for the future, and $\bar{N}_{pr}$ is estimated from climate scenarios. Following the assumption that $\bar{c}, G_{\widetilde{N}}, G_{\widetilde{c}\widetilde{N}}$ are constant for both periods,

$\bar{c}_r = \bar{c}_{pr}, G_{\widetilde{N}r} = G_{\widetilde{N}pr}, G_{\tilde{c}\widetilde{N}r} = G_{\tilde{c}\widetilde{N}pr}$ (a "basic parametrization scheme" according to Kovalenko, 1993); new parameters of the Pearson equation are calculated from $\bar{N}_{pr}$ :

$$a = \left(G_{\tilde{c}\widetilde{N}pr} + 2\bar{N}_{pr}\right)/\left(2\bar{c}_{pr}\right) , \tag{10}$$

$$b_0 = -G_{\widetilde{N}pr}/\left(2\bar{c}_{pr}\right) , \tag{11}$$

$$b_1 = G_{\tilde{c}\widetilde{N}pr}/\bar{c}_{pr} . \tag{12}$$

Finally, the non-central moments of runoff are calculated for the projected period (denoted by a low index *pr*):

$$m_{1pr} = a - b_1 , \tag{13}$$

$$m_{2pr} = -b_0 - 2m_{1pr}b_1 + am_{1pr} , \tag{14}$$

$$m_{3pr} = -2m_{1pr}b_0 - 3m_{2pr}b_1 + am_{2pr} . \tag{15}$$

It should be noted that in core version 0.2 the linear filter model includes the multiplicative stochastic component (see Annex 1 for details). It may leads to unstable solutions for the Fokker-Plank-Kolmogorov equation ($m_k \to \infty$) for statistical moments of high orders. Two methods for getting stable solutions for the Fokker-Plank-Kolmogorov equation are suggested by Kovalenko (2004), and one of them is already implemented in core version 0.1 (Shevnina et al., 2017).

### 2.4 Model output

In our study, the EPC of runoff was modelled within Pearson Type III distribution. This distribution is commonly used by water engineers to estimate water extremes (Kountrouvelis and Canavos, 1999; Rogdestvenskiy and Chebotarev, 1974; Matalas and Wallis, 1973). The water engineering guidelines provide the ordinates of EPCs from look-up tables (Guidelines, 1984) depending on the CV and CS. These coefficients are calculated from non-central moments' estimates (Rogdestvenskiy and Chebotarev, 1974):

$$CV = \sqrt{\left(m_2 - m_1^2\right)}/m_1 , \tag{16}$$

$$CS = \left(m_3 - 3m_2m_1 + 2m_1^3\right)/CV^3m_1^3 . \tag{17}$$

The MARCS$^{\text{HYDRO}}$ model output includes the estimates of the mean value, CV and CS, calculated for the reference period from observations as well as these estimates simulated from mean precipitation for the projected period. The ordinates of the EPC available from look-up tables then allows the calculation of the runoff values together with their exceedance probability.

### 3 Model application: A case study

In our study, we chose the basin of the Iijoki River at the Raasakka gauge (Lat 25.4º / Lon 65.3º) in order to give an example of the application of the MARCS$^{HYDRO}$ model on the catchment scale. The Iijoki River is located in north-west Finland, and the Raasakka gauge outlines a watershed area of over 14191 km$^2$. The catchment has a small population and there are no hydropower plants of multi-year regulation to affect the natural regime of the annual cycle. Thus, one can expect that historical yearly time series of the annual runoff rate do not contain trends connected to artificial regulation. This case study shows an example of the set-up and output of the probabilistic MARCS$^{HYDRO}$ model.

### 3.1 The MARCS$^{HYDRO}$ model set-up: The reference period

The yearly time series of volumetric water discharge of the Iijoki River were extracted from a dataset of the GRDC (GRDC 56068 Koblenz, Germany). The observations at the Raasakka gauge (ID = 6854600) cover the period 1911–2014, and they do not contain gaps. This period was considered as the reference period. The annual specific water discharge (ARR, mm year$^{-1}$) was calculated from the average volumetric water discharge for each year in the reference period. Then, the non-central moments were calculated from the yearly time series of the ARR with the method of moments (see Table 1). The reference climatology (the means of precipitation and air temperature) were evaluated from the dataset of NOAA (NOAA/OAR/ESRL PSD, Boulder, Colorado, USA) at a grid node nearest to the watershed centroid (this technique will be discussed in a separate paper, as will the methods of a forcing pre-analysis).

### 3.2 The MARCS$^{HYDRO}$ model forcing: The projected period

Climate scenarios provide a range of projections for temperature and moisture regimes in the future. This range is produced by different assumptions about climate scenarios as well as specific climate models. However, the climate projections include precipitation and air temperature, and they give a forcing to hydrological models in order to simulate projections of runoff. In the case study of the Iijoki River, the data from CMIP5 (Taylor et al., 2012) for three representative concentration pathways (RCPs) were used to force the MARCS$^{HYDRO}$ model. For each RCP scenario, the projections of annual precipitation rate were applied to test how the MARCS$^{HYDRO}$ model simulates the EPC under different forcing trajectories. For the period of 2020–2050 (considered the projected period), the mean values of the precipitation rate (mm year$^{-1}$) were calculated based on four world-leading global climate models. We used the outputs from the global models CaESM2 (Chylek et al., 2011), HadGEM2-ES (Collins et al., 2011), INM-CM4 (Volodin et al., 2010) and MPI-ESM-LR (Giorgetta et al., 2013) (see Table 2). The mean values of the precipitation rate varied by 2–5 % of the model's average over the RCP scenarios; however, these values alter substantially between the climate models. Among the outputs considered, the MPI-ESM-LR model projects the highest changes in the mean values of the precipitation rate compared to the reference period (see Tables 1 and 2). The HadGEM2-ES model gives the lowest values for the mean values of the precipitation rate. The projected means of the precipitation rate varied slightly between the scenarios. At the same time, they exhibited a significant range of changes

among the climate models (the mean values of the precipitation rate ranges from 619 to 737 mm year$^{-1}$) for the case of the
Iijoki River at Raasaka.

### 3.3 The MARCS$^{HYDRO}$ model output: The projected period

The projected non-central moments' estimates were simulated for the scenarios/models listed in Table 2. These estimates
were used to calculate the mean value, CV and CS (see Eq. (16–17)) that were included in the output of the MARCS$^{HYDRO}$
model. Table 3 shows the modelling results for the HadGEM2-ES and MPI-ESM-LR global models, where the water
discharges with 10 and 90 % exceedance probabilities are given. The ordinates of the Pearson Type III distribution were
extracted from the look-up tables used in hydrological engineering (Druzhinin and Sikan, 2001), and they allow expressing
runoff as water discharge (m$^3$s$^{-1}$). For the Iijoki River at Raasakka, the mean values of ARR and CV vary under the RCP
scenarios by over 7 % and 5 % correspondingly. The maximum alteration in the projected mean values of ARR was obtained
under RCP85 (619 to 737 mm year$^{-1}$). Under the projections of the MPI-ESM-LR model, the mean ARR increases by over 17 %.

In the case of the Iijoki River at Raasakka gauge, the 10 % water discharge exceedance probability will increase in the future under
the scenarios/models considered (see Table 3). It may leads to risks of energy spills at hydropower stations located within the
catchment of the Iijoki River in the period 2020–2050. At the same time, risks connected with water shortages may be fewer since
they are connected to a 90 % water discharge exceedance probability, which is predicted to increase. Figure 2 shows another way
in which the model performs the EPC of the annual runoff rate for the Kyrönjoki River at Skatila gauge (GRDC ID: 6854900). The
set of EPCs were simulated under three RCP scenarios using a similar set-up to the MARCS$^{HYDRO}$ model (Shevnina et al., 2019). In
the further development of the visualisation block, it would be important to involve water managers and decision makers in order to
better outline practical applications for the probabilistic hydrological model.

### 4 Discussions

Nowadays, the future vision of the climate is changing continuously. Climate projections are updated almost every 5–6 years and
many climate models generate meteorological projections for variables such as precipitation and air temperature. Hydrological
models are needed to perform an "express analysis" about future changes in water resources and water extremes (floods and
droughts) on a climate scale. The climate scale means that the express analysis is provided for a period of 20–30 years. Lumped or
semi-distributed physically based hydrological models are traditionally used on a short-term or seasonal scale to simulate a runoff
time series from a time series of meteorological variables (Seibert, 1999). In many catchment-scale hydrological studies, these
models are driven by the outputs of climate models or their ensembles in order to evaluate water resources and extremes in the near
future (Arheimer and Lindström, 2015; Veijalainen et al., 2012; Yip et al., 2012). The simulation of the runoff time series from a
time series of meteorological variables (see Fig. 2 in Veijalainen et al. [2012]) leads to high computational costs for such
estimations that need to be served in terms of probability in economical applications (Murphy, 1976). The probabilistic

MARCS[HYDRO] model is computationally cheaper when compared to lumped or semi-distributed physically based hydrological models. It can easily be coupled with global and regional climate models, and it can provide the express analysis of water resources under a modern version of the future climate.

In this paper we described the structure for the probabilistic hydrological MARCS[HYDRO] model, together with the AFA method that lies behind the new model's core version 0.2. The AFA method has a more than 25-year-long history; however, most of the studies are published in Russian (Kovalenko, 1993; 2004; 2009; Kovalenko et al., 2010). The AFA method is based on the statistical theory of automatic systems (Pugachev et al., 1974), which is an outsider among the "classical hydrological" disciplines. The AFA method is one simplification of the Fokker-Plank-Kolmogorov equation approach that has been developed in the Russian State Hydrometeorological University. It has been tested in many case studies on river basins located in Russia, Colombia, Bolivia, Mali etc. There are also a number of publications in English (Rosmann and Domínguez, 2017; Shevnina et al., 2017; Kovalenko, 2014; Domínguez and Rivera, 2010; Viktorova and Gromova, 2008). In this manuscript we formulated the theory logically in an attempt to provide the equations for the new core 0.2 of the MARCS[HYDRO] model; however, it also needs to describe the AFA method that lies behind it.

The probabilistic hydrological MARCS[HYDRO] model includes the core versions 0.1 and 0.2. In both cores, only three non-central moments are evaluated to construct the EPC within the theoretical distribution the Pearson III Type, which is among the traditional distributions of the frequency and risk analysis in hydrology (Kite, 1977; Rogdestvenskiy and Chebotarev, 1974; Sokolovskiy, 1968; Elderton, 1969; Benson, 1968). The model simulates three estimates of non-central moments of runoff instead of a runoff time series, and this circumstance makes the computations by the MARCS[HYDRO] model "low cost" compared to conceptual hydrological models (Arheimer and Lindström, 2015; Veijalainen et al., 2012). The MARCS[HYDRO] model allows putting the projections of runoff in terms of probability, that is, they appear as runoff values together with their exceedance probability.

The MARCS[HYDRO] model includes six modules, and each module allows improvements by including new methods. In this paper, the new model – core version 0.2, extended to simulate the third statistical estimator (skewness) – is presented. The applicability of core version 0.2 is limited by the assumptions behind the AFA approach. Among others, there is the "quasi-stationary" assumption for the expected climate change. In this case, the climate is described by the statistical estimators (i.e. mean value, variability etc.) of precipitation, air temperature, evapotranspiration, river runoff etc. for the period of 20–30 years. It is assumed to consider two time period periods with statistically different climates, namely the reference period and the projected period. Another limitation is connected to the linear filter stochastic model (for details, see Annex 1) used in core version 0.2. It should be noted that there is a multiplicative component in the model core, and it may lead to unstable solutions of the Fokker-Plank-Kolmogorov equation. Kovalenko (2004) suggests two solutions that result in stable solutions of the Fokker-Plank-Kolmogorov equation. One of the solutions was given by Kovalenko et al. (2010) and is coded in model version 0.1 (Shevnina et al., 2017). However, a checking procedure needs to be applied before using this core version. In the checking procedure we plan to use the "beta criterion" method suggested by Kovalenko (2004) to further develop the MARCS[HYDRO] model.

Further improvements of the MARCS$^{HYDRO}$ model are going to be implemented in the block of parametrization and hindcasts. Recently, only the basic parametrization scheme (Kovalenko, 1993) has been included. This basic scheme gives over 70–80 % successful hindcasts ("forecasts in the past") in the model cross-validation (Shevnina et al., 2017), and the implementation of a regionally oriented parametrization scheme (Shevnina, 2011) is the next step. It needs to include a mean value of the air temperature of the parameter, connected to "noised" watershed physiography in Eq. (A.4), the inverse of the runoff coefficient in the work of Kovalenko (1993). It is also important to study the role of the spatial resolution of meteorological forcing in affecting the modelling uncertainties for the simulated mean, CV and CS of runoff.

To fine the probabilistic MARCS$^{HYDRO}$ model among other hydrological models, its practical applications needs to be better outlined. The model serves a probabilistic form of long-term hydrological projections, and they require adaptation to the needs of water engineers and water managers as a tool for risk analysis under the expected climate change. The projected EPCs of multi-year river runoff can be applied in designing bridges, pipes, dams etc. in order to minimise the future risks connected to extreme floods (Shevnina et al., 2017; Kovalenko et al., 2014; Kovalenko, 2009) and to water shortage due to droughts (Viktorova and Gromova, 2014). It is important to define informative forms for the outputs of the MARCS$^{HYDRO}$ model that can be adapted to the needs of a practice, and the development of the block of economic application is among the others studies that are to be continued in close cooperation with water managers and decision makers.

## 5 Conclusions

The paper describes the theory and assumptions of the AFA approach, as well as the probabilistic hydrological MARCS$^{HYDRO}$ model's structure and core version 0.2. The features of the model are: the close connection to water engineering due to serving the runoff projection in terms of probability, cheapness in terms of computational cost and a wide range of techniques allowing model improvement. In the new core, the third moment linked to the location parameter of the Pearson Type III distribution (or asymmetry) was implemented for simulation. In the previous version of the model core, a constant CS/CV ratio is used to calculate the location parameter of the distribution.

To give a practical example how to set up the MARCS$^{HYDRO}$ model, the case of the Iijoki River at Raasakka located in Finland was considered. The model simulated the tailed values of 10 % and 90 % of annual water discharge from the outputs of global climate models. We showed two forms of the probabilistic projections of runoff: an EPC and the runoff values with their exceedance probability. This case study of the Iijoki River at Raasakka shows that the MARCS$^{HYDRO}$ model gives reasonable results for the meteorological projections considered. The practical applications in water management and decision-making should be clarified in further studies in close co-operation with water engineers.

**6 Code availability**

Currently, the MARCS<sup>HYDRO</sup> model code is hosted at https://github.com/ElenaShe000/MARCS, with details of its applications for catchment-scale case studies. The model source code for core version 0.2 is distributed under the Creative Commons Attribution 4.0 License and can be downloaded from the link https://zenodo.org/record/1220096#.WyTXxxxRVhw and used freely in scientific research with reference to this publication. We hope that this type of license provides the best way to create a community of motivated people to further develop the model. Then, the source code will be distributed under the terms of a user agreement.

**7 Data availability**

The following data sets can be used to set up and force the MARCS model: the GRDC (GRDC, 56068 Koblenz, Germany), the NOAA/OAR/ESRL PSD (Boulder, Colorado, USA) and CMIP5 (Taylor et al., 2012).

**8 Sample availability**

The sample data set for the case study of the Iijoki river at Raasaka is given at this site: 310 https://zenodo.org/record/1220096#.WyTXxxxRVhw.

**Annex 1. The theoretical basis for core version 0.2**

**A1.1 The assumptions behind advance of frequency analysis**

Advance of frequency analysis (AFA) is based on the theory of stochastic systems, specifically, the Fokker-Plank-Kolmogorov equation, which is simplified into a system for three non-central statistical moments (Pugachev et al., 1974). 315 The time series of annual runoff is considered as a realisation of a random-process Markov chain type that is assumed to be "stationary". It means that the statistical estimators (mean, variance and skewness) do not change over the period considered. The statistical estimators are used to model an exceedance probability curve (EPC) of the annual runoff with Pearson Type III distribution. The AFA approach is developed with an assumption of "quasi-stationary" (Kovalenko et al., 2010, Kovalenko, 1993). The quasi-stationary assumption suggests that the statistical estimators of multi-year runoff are different 320 for two periods (the reference period and the projected period). For the reference period, the statistical estimators are evaluated from historical yearly time series of runoff. For the projected period, the statistical estimators of runoff are simulated based on the outputs of global- or regional-scale climate models under any climate scenario.

**A1.2 The linear filter stochastic model**

In this context, models replace a complicate hydrological system using maths abstractions and aim to reveal the spatial and 325 temporal runoff features which are important depending on the goals of study. Among other models, "black box"

hydrological models consider a river basin as a dynamic system with lumped parameters. These models are "based on analysis of concurrent inputs and temporal output series" (WMO-№168, 2009) and transform series of meteorological variables (precipitation, air temperature) into series of runoff. Both input and output series are functions of time (WMO-№168, 2009):

$$a_n(t)\frac{d^n Q}{dt^n}+a_{n-1}(t)\frac{d^{n-1}Q}{dt^{n-1}}+...+a_1(t)\frac{dQ}{dt}+a_0(t)Q=$$
$$=b_n(t)\frac{d^n P}{dt^n}+b_{n-1}(t)\frac{d^{n-1}P}{dt^{n-1}}+...+b_1(t)\frac{dP}{dt}+b_0(t)P \tag{A.1}$$

where $Q$ is the runoff in volumetric flow rate, $P$ is the precipitation in volumetric flow rate (rain, snow melt) and the coefficients $a_i$ and $b_i$ are the empirical parameters of a translating system. These coefficients are the lumped parameters of the black box model. The solution to Eq. (A.1) for zero initial conditions gives:

$$Q(t)=\int_0^t h(t,\tau)P(\tau)d\tau \tag{A.2}$$

where the function $h(t,\tau)$ represents the response of a river basin at time $t$ to a single portion of precipitation at time $P$. In the AFA approach, a river basin is considered as a linear system, transforming the annual precipitation into the annual runoff:

$$a_1(t)\frac{dQ}{dt}+a_0(t)Q=b_0(t)P . \tag{A.3}$$

On the other hand, a river basin can be considered as a linear system with stochastic components in the input function and the model parameter (Kovalenko, 1993):

$$dQ=[-(\bar{c}+\widetilde{c}(t))Q+(\bar{N}+\widetilde{N}(t))]dt , \tag{A.4}$$

where $a_0(t)=\bar{c}+\widetilde{c}(t)$ is the stochastic parameter of the system (a "noised" watershed physiography, the inverse of runoff coefficient), $b_0(t)P=\bar{N}+\widetilde{N}(t)$ is the stochastic input for the system (a "noised" precipitation), and $a_1=1$. The stochastic components of $\widetilde{c}(t)$ and $\widetilde{N}(t)$ are the Gaussian "white noise" with zero means, and their intensities are $G_{\widetilde{c}}$, $G_{\widetilde{N}}$. The intensities are mutually correlated as $K_{\widetilde{c}\widetilde{N}}(\tau)=E(\widetilde{c}(t)\widetilde{N}(t+\tau))=G_{\widetilde{c}\widetilde{N}}\delta(\tau)$. It should be noted, that the multiplicative parameter $\bar{c}+\widetilde{c}(t)$ in Eq. (A.4) is the sum of the constant $\bar{c}$ and Gaussian «white noise» $\widetilde{c}(t)$, and it may lead to the unstable solutions of the Fokker-Plank-Kolmogorov equation (i.e. in may lead to infinite statistical moments of high orders). It limits the application of the AFA method (Kovalenko, 1993). Kovalenko (2004) suggests two solutions, and we will introduce them in a further paper.

## A1.3 The Fokker-Plank-Kolmogorov equation and simplifications

The Fokker-Plank-Kolmogorov equation can be applied to simulate the probability density function (PDF) for the stochastic $Q(t)$ in Eq. (4) (Kovalenko, 1993; Pugachev, 1974):

$$\frac{\partial p(Q,t)}{\partial t} = -\frac{\partial}{\partial Q}\left(A(Q)\, p(Q,t)\right) + 0.5\frac{\partial^2}{\partial Q^2}\left(B(Q)\, p(Q,t)\right) , \tag{A.5}$$

where $p(Q,t)$ is the PDF of $Q$ at time $t$; and the drift coefficient ($A(Q)$) and diffusion coefficients ($B(Q)$) are calculated as follows (Kovalenko, 1993; Pugachev, 1974):

$$A(Q) = -\left(\bar{c} - 0.5 G_{\widetilde{c}}\right)Q - 0.5 G_{\widetilde{c}\,\widetilde{N}} + \bar{N} , \tag{A.6}$$

$$B(Q) = G_{\widetilde{c}}\, Q^2 - 2 Q G_{\widetilde{c}\,\widetilde{N}} + G_{\widetilde{N}} . \tag{A.7}$$

The analytical solution of Eq. (A.5) is difficult and not always needed for practical applications in water engineering since the PDFs of runoff are modelled from a set of statistical estimators, and the moments are from, among others, van Gelder et al. (2006). The PDFs are described with the set of moments $m_k = \int_{-\infty}^{+\infty} Q^k p(Q,t)\, dQ$ (where $k$ is number of the moment, $k \to$

$\infty$). To obtain the equations for $m_k$, both sides of Eq. (A.5) were multiplied by a differentiable function $\psi(Y)$ and then integrated within limits from $-\infty$ to $+\infty$ by $Q$ (however, it is supposed that $Q > 0$):

$$\frac{d\left(\int_{-\infty}^{+\infty} \psi(Q)\, p(Q,t)\, dQ\right)}{dt} = \int_{-\infty}^{+\infty} p(Q,t)\, A(Q)\frac{\partial \psi(Q)}{\partial Q}\, dQ + 0.5\int_{-\infty}^{+\infty} p(Q,t)\, B(Q)\frac{\partial^2 \psi(Q)}{\partial Q^2}\, dQ \tag{A.8}.$$

Then, $\psi(Q)$ was replaced with $\psi(Q) = Q^k$, and Eq. (A.8) was written as:

$$\frac{dm_k(t)}{dt} = \int_{-\infty}^{+\infty} p(Q,t)\, A(Q)\frac{\partial\left(Q^k\right)}{\partial Q}\, dQ + 0.5\int_{-\infty}^{+\infty} p(Q,t)\, B(Q)\frac{\partial^2\left(Q^k\right)}{\partial Q^2}\, dQ . \tag{A.9}$$

For a stationary random process $dm_k(t)/dt = 0$, and the drift and diffusion coefficients are constant. Thus, Eq. (A.9) was simplified as follows:

For $k=1$:

$$-\left(\bar{c} - 0.5 G_{\widetilde{c}}\right)m_1 - 0.5 G_{\widetilde{c}\,\widetilde{N}} + \bar{N} = 0 . \tag{A.10}$$

For $k \geq 2$:

$$-k\left(\bar{c} - 0.5 k G_{\widetilde{c}}\right)m_k + k\bar{N} m_{k-1} - k(k-0.5)G_{\widetilde{c}\,\widetilde{N}} m_{k-1} + 0.5 k(k-1)G_{\widetilde{N}} m_{k-2} = 0 . \tag{A.11}$$

Further, the summands in Eq. (10–11) were divided by $\left(2\bar{c} + G_{\widetilde{c}}\right)$, and new notations were introduced as suggested in the work of Kovalenko (1993) and Pugachev et al. (1974):

$$a=\frac{G_{\widetilde{c}\widetilde{N}}+2\bar{N}}{2\bar{c}+G_{\widetilde{c}}} \;\;;\;\; b_0=-\frac{G_{\widetilde{N}}}{2\bar{c}+G_{\widetilde{c}}} \;\;;\;\; b_1=\frac{2G_{\widetilde{c}\widetilde{N}}}{2\bar{c}+G_{\widetilde{c}}} \;\;;\;\; b_2=-\frac{G_{\widetilde{c}}}{2\bar{c}+G_{\widetilde{c}}} \;.$$

Then, for $k = 1, 2, 3, 4$ the system of Eq. (A.10–11) includes:

$$m_1\left(2b_2+1\right)-a+b_1=0 \;, \tag{A.12}$$

$$\left(3b_2+1\right)m_2+\left(2b_1-a\right)m_1+b_0=0 \;, \tag{A.13}$$

$$\left(4b_2+1\right)m_3+\left(3b_1-a\right)m_2+2b_0 m_1=0 \;, \tag{A.14}$$

$$\left(5b_2+1\right)m_4+\left(4b_1-a\right)m_3+3b_0 m_2=0 \;. \tag{A.15}$$

The set of four moments $(m_1, m_2, m_3, m_4)$ is sufficient to model distributions from the Pearson equation (Andreev et al., 2005; Elderton and Johnson, 1969). However, in water engineering we usually only use three-parametric probability distributions fitted to observations (Guidelines, 2004; Guidelines, 1984; Bulletin 17-B, 1982). In this case, $G_{\widetilde{c}} << \bar{c}$ is assumed, thus it leads to $b_2=-G_{\widetilde{c}}/\left(2\bar{c}+G_{\widetilde{c}}\right)\approx 0$ and $\left(4b_2+1\right)\approx 1$ , $\left(3b_2+1\right)\approx 1$ , $\left(2b_2+1\right)\approx 1$ . To model the PDFs (or EPCs) of annual runoff within the Pearson Type III distribution, the system of Eq. (A.12–15) is simplified as follows:

$$-a+b_1=-m_1 \;, \tag{A.16}$$

$$b_0+2m_1 b_1-am_1=-m_2 \;, \tag{A.17}$$

$$2m_1 b_0+3m_2 b_1-am_2=-m_3 \;. \tag{A.18}$$

Denoting $lk=\begin{pmatrix}-m_1\\-m_2\\-m_3\end{pmatrix}$ , $x=\begin{pmatrix}b_1\\b_0\\a\end{pmatrix}$ and $A=\begin{pmatrix}1 & 0 & -1\\2m_1 & 1 & -m_1\\3m_2 & 2m_1 & -m_2\end{pmatrix}$ , the parameters $a, b_0, b_1$ are calculated as $x_i = D_i / D$, where $D$ is the determinant of matrix $A$, and $D_i$ is the determinant of the matrix obtained by replacing of the column $i$ (1, 2,

3) in matrix $A$ by the vector $lk$. Finally, the parameters $a, b_0, b_1$ are calculated as follows:

$$b_1=0.5\left(3m_1 m_2-2m_1^3-m_3\right)/\left(m_2-m_1^2\right) \;, \tag{A.19}$$

$$b_0=0.5\left(m_1^2 m_2-2m_2^2+m_1 m_3\right)/\left(m_2-m_1^2\right) \;, \tag{A.20}$$

$$a=0.5\left(5m_1 m_2-4m_1^3-m_3\right)/\left(m_2-m_1^2\right) \;. \tag{A.21}$$

### A1.3 Notations

There are too many notations used to describe the model's core version 0.2, thus the secondary parameters of equations were grouped by the model behind it. Table A.1 shows the notation and description of the secondary parameters for the linear filter

stochastic model. Eq. (A.3) is a simplification of Eq. (A.1) that limits the first order ordinal differential equation. It includes three parameters, $a_0$, $a_1$ and $b_0$, and two of them are assumed to be noised. These noised parameters include a constant component (indicated with a bar) and a Gaussian white noise component (indicated with a tilde) with their own intensities.

Table A.1 The notation and description of the parameters for a linear filter stochastic model.

| $Q$ | runoff as a volumetric flow rate, $m^3s^{-1}$ |
|---|---|
| $P$ | precipitation as a volumetric flow rate, $m^3s^{-1}$ |
| $a_i(t)$, $b_i(t)$ | the lumped parameters of "block box" model, $i = 0$ and 1 |
| $\bar{c} + \tilde{c}(t)$ | the inverse of runoff coefficient: $\bar{c}$ is constant component, $\tilde{c}(t)$ is the Gaussian "white noise" |
| $\bar{N} + \tilde{N}(t)$ | precipitation: $\bar{N}$ is constant component, $\tilde{N}(t)$ is the Gaussian "white noise" |
| $G_{\tilde{c}}$, $G_{\tilde{N}}$ | the intensities of the Gaussian "white noise" |
| $G_{\tilde{c}\tilde{N}}\delta(\tau)$ | the correlation function for the mutually delta-correlated processes $\tilde{c}(t)$ and $\tilde{N}(t)$ |

Table A.2 gives a description of the parameters of the Fokker-Plank-Kolmogorov equation and the Pearson system. It should be noted that we do not solve the Fokker-Plank-Kolmogorov equation, and only its simplification for the system of three non-central moments is applied. These non-central moments are estimated from runoff observations for the reference period. For the projected period the moments are calculated from the mean of precipitation.

Table A.2. The notations of the Fokker-Plank-Kolmogorov equation and Pearson equation.

| $p(Q,t)$ | the probability density function of $Q$ at time $t$ |
|---|---|
| $A(Q)$ | the drift coefficient, estimated from the "noised" parameters and their intensities |
| $B(Q)$ | the diffusion coefficient, estimated from the "noised" parameters and their intensities |
| $m_k$ | the non-central statistical moment with order $k = 1, 2, 3, 4$ |
| $a, b_0, b_1, b_2$ | the parameters of a distribution within the Pearson equation |

**Annex 2. A short user guide the MARCS model**

To set up the model for a single river catchment, the non-central moments should be calculated from historical time series of the annual river runoff rate as well as from a mean value of annual precipitation rate. These values should be placed manually (lines 45–48 in model_core.py located at https://zenodo.org/record/1220096#.WyTXxxxRVhw) as should the ID number of catchment (line 51 of model_core.py). To force the model, the projected mean value of the annual precipitation rate should be evaluated from an output of a climate model, and then the model_core.py can be run in the Unix command line: ./model_core.py XXX (where

XXX is the mean of the annual precipitation rate for the projected period). The output of model_core.py is stored in the output file model_GPSCH.txt and included in line with the following format: the ID of catchment, the first non-central moment estimate of annual runoff rate (mm year[-1]) for a reference period, the mean value of annual precipitation rate (mm year[-1]) for a reference period,

the coefficient of variation for a reference period, the coefficient of skewness for a reference period, the model parameters $\bar{c}$, $G_{\widetilde{N}}$, $G_{\widetilde{c}\,\widetilde{N}}$, the the first non-central moment estimate of annual runoff rate (mm year[-1]) for a projected period, the mean value of annual precipitation rate (mm year[-1]) for a projected period, the coefficient of variation for a projected period, the coefficient of skewness for a projected period.

**Competing interests**

The authors declare that they have no conflict of interests.

**Authors' contribution**

Elena Shevina contributes to the MARCS[HYDRO] model coding and to the Russian-English translation of some parts from Kovalenko (1993), Kovalenko (2004) and Pugachev et al. (1974). She is responsible for writing the text . Andrey Silaev supports the theoretical part of the AFA method (Pugachev et al. 1974; Kovalenko; 1993) and formulates the equations step

by step. Both authors care for the terms used in the paper, which we have been trying to write on " a language in common" between water engineering and radiophysics.

**Acknowledgement**

The study is supported by the Academy of Finland (contract 283101). Authors would thank two anonymous referees, editor and A. Frolov for their comments. Our special thanks to Alexander Krasikov, who supports the model coding.

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

**Table 1.** The MARCS$^{\text{HYDRO}}$ model set-up: the Iijoki river at Raasakka as a case study.

| GRDC ID | River at Gauge | Length, year | $m_{1\,r}$, mm year$^{-1}$ | $m_{2\,r}$, mm$^2$ year$^{-1}$ | $m_{3\,r}$, mm$^3$ year$^{-1}$ | $\bar{N}_r$, mm year$^{-1}$ | $\bar{T}_r$ $^*$, °C |
|---|---|---|---|---|---|---|---|
| 6854600 | Iijoki at Raasakka (Finland) | 100 | 379 | 149343 | 60811610 | 625 | 0.2 |

Notes: $m_{1r}, m_{2r}, m_{3r}$ are the moments of runoff as well as the mean of precipitation ( $\bar{N}_r$ ) were evaluated from observations. The mean air temperature ( $\bar{T}_r$ )$^*$ was not used in the model set up in case of the Iijoki River, however this value allows advancement of the model parametrization (Shevnina et al., 2017).

**Table 2.** The forcing of the MARCS[HYDRO] model for the case study of the Iijoki River at Raasakka.

| Global climate model | Climate scenario | | | | | |
|---|---|---|---|---|---|---|
| | RCP26 | | RCP45 | | RCP85 | |
| | $\bar{T}_{pr}$, °C* | $\bar{N}_{pr}$, mm year$^{-1}$ | $\bar{T}_{pr}$, °C | $\bar{N}_{pr}$, mm year$^{-1}$ | $\bar{T}_{pr}$, °C | $\bar{N}_{pr}$, mm year$^{-1}$ |
| CaESM2 | 2.9 | 673 | 2.7 | 652 | 2.7 | 652 |
| HadGEM2-ES | 1.4 | 635 | 2.6 | 637 | 2.2 | 619 |
| INM-CM4 | – | – | 1.3 | 645 | 1.4 | 660 |
| MPI-ESM-LR | 2.5 | 704 | 2.2 | 695 | 2.9 | 737 |

Notes: Projected mean of air temperature ( $\bar{T}_{pr}$ )* is needed for a regional parametrization scheme (see details Shevnina, 2011), and these values were not used in the model forcing in the case of the Iijoki River at Raasakka. $\bar{N}_{pr}$ is the projected

mean of annual precipitation amount.

**Table 3.** The projected climatology and statistics of annual runoff: a case of the Iijoki River.

| Value | Reference period: 1914–2014 | Projected period: 2020–2050 | | | | | |
|---|---|---|---|---|---|---|---|
| | | HadGEM2-ES | | | MPI-ESM-LR | | |
| | | RCP85 | RCP45 | RCP26 | RCP85 | RCP45 | RCP26 |
| Precipitation, mm year$^{-1}$ | 625 | 619 | 637 | 635 | 737 | 695 | 704 |
| Specific discharge, mm year$^{-1}$ | 380 | 375 | 386 | 385 | 447 | 421 | 427 |
| CV | 0.19 | 0.2 | 0.19 | 0.19 | 0.16 | 0.17 | 0.17 |
| CS | –0.04 | –0.04 | –0.04 | –0.04 | –0.04 | –0.04 | –0.04 |
| $Q_{10\%}$, m$^3$s$^{-1}$ | 475 | 473 | 483 | 481 | 527 | 505 | 512 |
| $Q_{90\%}$, m$^3$s$^{-1}$ | 293 | 278 | 297 | 296 | 331 | 354 | 359 |

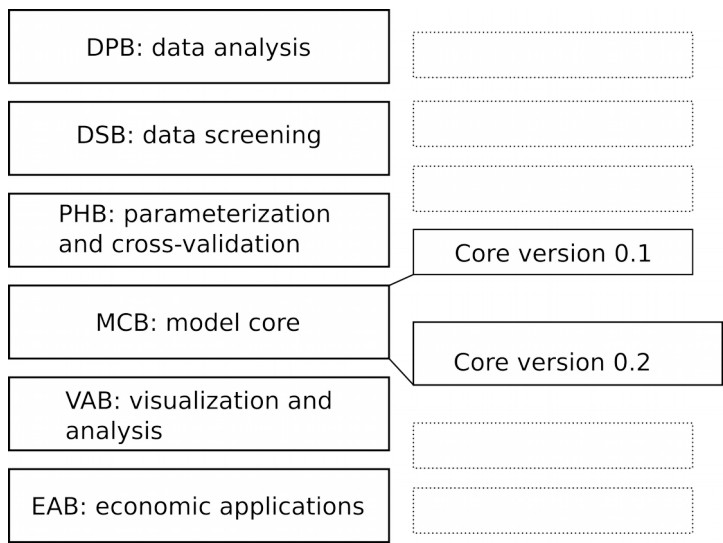

**Figure 1:** The MARCS<sup>HYDRO</sup> model structure and core versions.

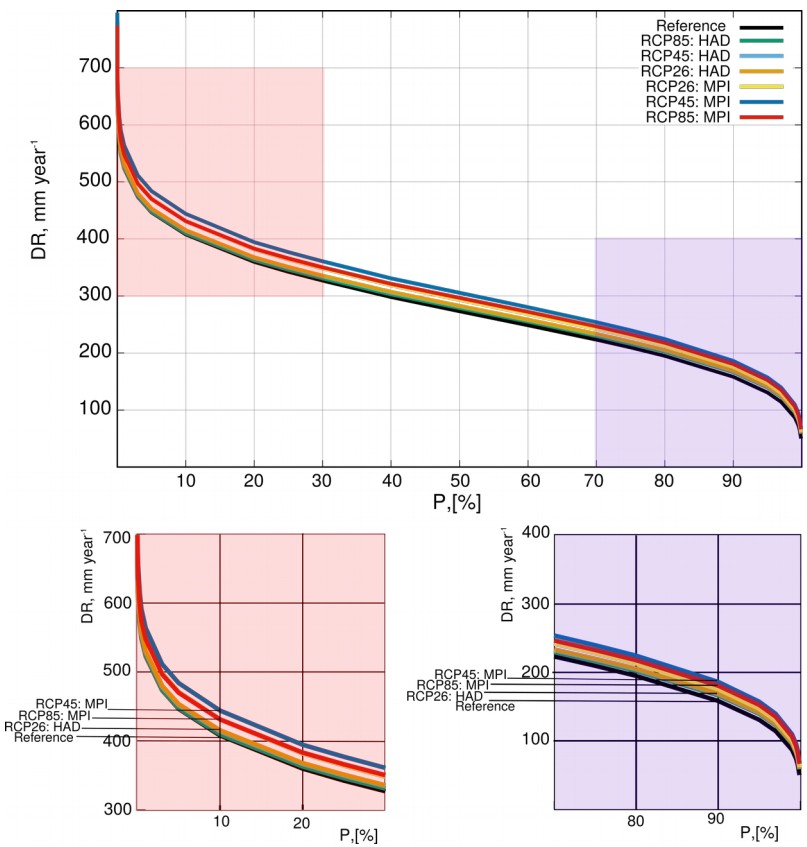

600

**Figure 2:** The variability of the tails of the EPCs for annual runoff for the reference period (black) and projected period (colours).