# Peer review of "The probabilistic hydrological MARCSHYDRO (the MARkov Chain System) model: Its structure and core version 0.2"

_Geoscientific Model Development, 2018_

## Short Comment (SC1) · 5 Aug 2018

Please, see supplement.

Please also note the supplement to this comment:
https://www.geosci-model-dev-discuss.net/gmd-2018-108/gmd-2018-108-SC1-supplement.zip

---

## Author Comment (AC1) · 15 Aug 2018

To the conclusion given by A. Frolov It should be noted, that most comment are actually not addressed to this particular study, which is the model description paper (MARCS model core version 0.2). Thus, it is not possible to discuss the Fokker-Plank-Kolmogorov approach in hydrology withing one particular paper. However, we are going to continue our study in the future. List with the answers by E.Shevnina and A.Silaev to the comments given by A.Frolov is included as Supplement.

Please also note the supplement to this comment:

[Figure]

https://www.geosci-model-dev-discuss.net/gmd-2018-108/gmd-2018-108-AC1-supplement.pdf

**Supplement:**

**Answers to comments and conclusion given by A. Frolov**

To answer to the comments given by lead scientist researcher of Water Problems Institute of Russian Academy of Sciences Dr.Sci. A. Frolov the authors will follow to the order in his list.

*To the comment 1 by Frolov A.:*

The "linear black box" model with stochastic components is used as core equation for the hydrological model because it fit an assumption that the multi-annual runoff time series can be considered as a simple Markov chain stationary process (Kovalenko, 1993). To proof this assumption in hydrology, an autocorrelation coefficient (AC) is usually applied (Dobrovolski, 2011; Rozdestvenskiy and Chebotarev, 1974), and the AC is statistically significant on numerous observed annual runoff time series (Ratkovich, 1993; Rogdestvenskiy, 1986; Ratkovich, 1976). In our study, the "linear black box" model was chosen as physical core to describe the annual river surface runoff, while the linear model of the river basin with a stochastic input process by Klemeš (1978) is mostly applied on seasonal and short-term predictions, i.e. it considers more complicate physical processes going on a river catchments.

To clarify this issue, the text above was added to the manuscript together with the following references:

Dobrovolskiy S.: Global change on river runoff. Moscow: GEOS, 660 pp., 2011.(in Russian).

Klemeš V.: Phisically based stochastic hydrologic analysis// Adv. in Hydroscience, 11, 285-386, 1978.

Ratkovich D.: Multi-year fluctuation of river surface runoff. Regularity and regulation. Leningrad, Gidrometizdat, 225 pp., 1976. (in Russian).

Ratkovich D.: Hydrological basis of water supply. Moscow, The Water Problem Institute of Russian Academy of Science, 429 pp., 1993. (in Russian).

Rogdestvenskiy A. V.: Spatial and temporal variations of river flow in USSR, Leningrad, Gidrometeizdat, 385 pp., 1988. (in Russian).

Generally, it is important continuously expand a range of hydrological models to be apply in various situations depending on time and space scales of a processes considered withing the framework of the particular study.

*To the comment 2 by Frolov A.:*

In our paper, the Eq. (4) do not include the parameter $k$ (the runoff coefficient, following Kovalenko, 1993), and the parameter $a_0$ (page 4, line 105,) includes the stochastic component of the Gaussian "white noise" with zero means. The Eq. (4) is written for a continue time, and the only single correlation function as well as the only single the coefficient of correlation are existed. While the time is discrete in practical calculations, there are different assumptions how the continue time can be presented discretely. In our study, the only stationary processes were considered (see page 5, line 126), thus the correlation functions and time relaxation issues do not considered. The revised manuscript includes this explanation in the section of Discussion. Since, the second comment given by A. Frolov is addressed more to Kovalenko (1993) we decided to include the critical discussion (in Russian) the AFA method used in Shevnina (2015) as a supplement to this manuscript.

*To the comment 3 given by A. Frolov:*

While the Eq. (4) allows the infinite values of non-central moments for combination of the parameters, the probability density function of annual runoff was assumed to be the functions of only three non-central moments in our study. This assumption is proofed by numerous studies addressed to the annual runoff distribution fitting to multi-year surface river runoff observations including time series of the annual runoff (van Gelder et al., 2006; Rogdestvenskiy, 1986; Matalas and Wallis, 1973). The distribution function of only three parameters are common in the

hydrological engineering practice (SP, 2004; Guidelines, 1986), and it was also used withing this particular study. To answer this comment the text above was added to the revised version on the manuscript in the section of Discussion. The numeration of the equation was also changed (since it was noticed that the Eq. 16 does not exist).

***To the comment 4 given by A. Frolov:***

In our model description paper we would like to rise the discussion on the Fokker-Plank-Kolmogorov approach in hydrology (Rosmann and Domínguez, 2017; Shevnina et al., 2017; Kovalenko et al., 2010; Domínguez and Rivera, 2010; Kovalenko, 1993) to define its place among others modeling approaches. In our manuscript we consider only the method of AFA, and we would like clarify any concrete critics, questions and comments connected to the MARCS model core version 0.2. The MARSC model is under development, and it includes many blocks (Shevnina, 2015). The results of the model validation together with details about the cross-validation procedure are previously presented by Shevnina et al. (2017) and Kovalenko (1993).

***To the comment 5 given by A. Frolov:***

Unfortunately, it is not directly mentioned why "the limits of integration are incorrectly indicated".

We can just suppose that this comment linked to the value of Q>0, thus $\int_{-\infty}^{+\infty} p(Q,t)$ may be rewrite

as $\int_{0}^{+\infty} p(Q,t)$ since Q>0. The Eq. (4) allows the $p(Q,t)>0$ for the Q<0. However, this situation can be recognized on a data preprocessing analysis (see details in lines 55-60, p. 6), thus it allows to modify the basic core (the Eq.(4) in this study) to more advanced. It could be the topic of other model description paper. The above text is now added to the section of Discussion.

***To the conclusion given by A. Frolov:***

It should be noted, that most comment are actually not addressed to this particular study, which is the model description paper. Thus, it is not possible to discuss the Fokker-Plank-Kolmogorov approach in hydrology withing one particular paper. However, we are going to continue our study in the future.

With the best regards
Elena Shevnina and Andrey Silaev

---

## Referee Comment (RC1) · Anonymous Referee #1 · 8 Nov 2018

**The probabilistic hydrological model MARCS (MARkov Chain System): the theoretical basis for the core version 0.2 (Shevnina and Silaev, 2018)**

**General comments**

At a first glance, the paper show how competent the authors are in probabilistic hydrological models. Reviewer thinks that key aspects of this research are to provide the theoretical background of Markov Chain System. The manuscript is well written and logically structured. The extensive literature review is much appreciated as well.

Even though the goal of the paper relies on the scope of GMD, the intuition of the approach is not clearly stated. Since the approach uses Markov chain system, for the recent scientific community, it may not be new. So, the reviewer suggests laying the objective of the paper in different way. Indeed, the authors showed much effort on the topic but there is not much about the model use and its description. The manuscript mentions the version of model is 2. Reviewer does not see properly how they are different. The assumptions of the model are not clearly stated. The paper is mathematically enriched. Sometimes, reader may lose the concentration due to inappropriate description of the technical jargons.

The conclusion made in this manuscript seems to be the summary of the whole content. It may need revision posing future research and recommendation of this research. Right now the direction of this research is not clear. The reviewer suggests including some potential application beyond the water engineering even though the method is similar to Pearson type distribution. The extension of the paper will be better if the idea of posing such approach in space. Such statement shall be made clearly.

At this moment, the effort of authors is appreciated but still needs further improvement as described below prior to acceptance.

**Constructive suggestions**

- Author mentioned three statistical moments in line 79. But these are not listed here. For general audience, reviewer suggests to list them.
- Section 1.1 is very rich in mathematical expression. Only audience or practitioner with sound mathematical background easily understands. But for general audience, this section shall be revised in a simpler way...
- Please briefly mention what kind of parameters are lumped one and why such is called.
- The reviewer wants to have implicit explanation of the secondary parameters like a, b, c and c, Gs. It is not clear how such empirical equations are related with either data or physics.
- Are the time-series data are daily or monthly or yearly as mentioned in line 227? It would be better to define the time scale.
- In order to make the paper strong, reviewer suggests having some key statistics pictorially. This means how the observed set and models are correlated. What is the degree of performance?
- Reviewer feels the paper is somewhat incomplete as in the several statements; the authors did not mention how future works will be proceed. They just envisioned about the future paper.

**Specific comments**

- In line 90, comma is needed between features and which.
- The authors mentioned in parenthesis ("the reference"). What does it mean? It seems the authors forgot to have proper citation. In line 158.
- There are three graphs in the paper however, they are not proper captions. In line 273 has Figure 2, but where is Figure 2?

---

## Author Comment (AC2) · 29 Nov 2018

**Answers to the potential review of the Anonymous Referee #1**
by Elena Shevnina and Andrey Silaev

**The Referee #1 concluded that the manuscript "still needs further improvement as described below prior to acceptance".**

**General comments:**

Comment: At a first glance, the paper show how competent the authors are in probabilistic hydrological models. Reviewer thinks that key aspects of this research are to provide the theoretical background of Markov Chain System. The manuscript is well written and logically structured. The extensive literature review is much appreciated as well.

*Answer: One formulates more precisely, the manuscript provides the basics of the statistical theory of automatic system (Pugachev et al., 1974), the simplifications behind to the Advance of Frequency Analysis (Kovalenko, 1993) as well as the equations used on the core version 0.2 of the probabilistic hydrological model MARCS (MARkov Chain System). In this manuscript, the authors not only translate the parts of two books with theoretical basis from Russian, but also try to formulate material logically and to provide the equations for the new core of the hydrological model. The theory of Markov Chain System is outside of the manuscript content, even it gives the name for the hydrological model with simple Markov Chain core (see the Eq. 2 in the manuscript). Only a couple of months ago, the authors realized that the model with name MARCS is already exists (http://marcs.astro.uu.se/index.php), however the official name is the MARCS – atmospheres. We would need to change the name of our probabilistic hydrological model (MARCS) or becomes to be involved to the MARCS model community with the probabilistic hydrological model MARCS – hydrosphere. It would probably needs to change the version of the core, status and content of the code during revision process.*

*These comments are incorporated to the revision of the manuscript.*
*Pugachev, V.S., Kazakov, I.E. and Evlanov, L.G.: Basics of statistical theory of automatic system, Mashinostroenie, Moscow, USSR, 1974. (In Russian).*
*Kovalenko, V. V.: Modelling of hydrological processes, Gidrometeizdat, St. Petersburg, Russia, 1993. (In Russian).*

Comment: Even though the goal of the paper relies on the scope of GMD, the intuition of the approach is not clearly stated. Since the approach uses Markov chain system, for the recent scientific community, it may not be new. So, the reviewer suggests laying the objective of the paper in different way. Indeed, the authors showed much effort on the topic but there is not much about the model use and its description. The manuscript mentions the version of model is 2. Reviewer does not see properly how they are different. The assumptions of the model are not clearly stated. The paper is mathematically enriched. Sometimes, reader may lose the concentration due to inappropriate description of the technical jargons.

*Answer: We agree that the Markov Chain System approach is known by the recent scientific community and it is not a new. However, in this manuscript we attempted to explain the method used in the math "language", not on the intuition "language". On both "languages" it is not easy if the topic is on a boundary of two scientific disciplines (Hydrology and Statistical Radiophysics in our case). On the boundary, the therms may come from both sides to add or to complement each others, and it results to a specific jargon, which is noticed by the Referee 1. The back ground of the authors comes from the Hydrology (the frequency analysis and physical modeling) and the Radiophysics (the statistical theory of automatic system), and the explanations in the manuscript were given on the "language" in common. In our manuscript, we try to use the math equations as*

*much as possible to prevent non-correct description of the method due to the difference in the therms. It results to the "mathematically enriched text". In this manuscript, the core version 0.2 was presented in details. The previous model version 0.1 is shortly described in the Annex to Shevnina et al. (2017) without any theoretical details, which we have promised to present in our next manuscript. To follow our promises, this manuscript fills the gap and provides the theoretical basis of the probabilistic hydrological model MARCS. In the revised manuscript we stressed these two circumstances.*

*Shevnina, E., Kourzeneva, E., Kovalenko, V., and Vihma, T., 2017: Assessment of extreme flood events in a changing climate for a long-term planning of socio-economic infrastructure in the Russian Arctic, Hydrol. Earth Syst. Sci., 21, 2559-2578, doi:10.5194/hess-21-2559-2017.*

Comment: The conclusion made in this manuscript seems to be the summary of the whole content. It may need revision posing future research and recommendation of this research. Right now the direction of this research is not clear. The reviewer suggests including some potential application beyond the water engineering even though the method is similar to Pearson type distribution. The extension of the paper will be better if the idea of posing such approach in space. Such statement shall be made clearly.

*Answer: We agree, that it is important to place the probabilistic approach amount others hydrological modeling approaches. The general view on this place is done in Shevnina et al., 2017 (Fig. 1), and the details are provided in Shevnina et al. (2018). In the revised version of the manuscript we add one figure to show how different scientific disciplines are overlap in the AFA approach. It should be noted, that the statistical theory of automatic system is adopted to be used for a seasonal prediction of water inflow to hydropower reservoirs by Domínguez and Rivera (2010) and Shevnina (2001). There are also more studies published in Russian whose not included to the list of References since it is already long. It does not include a number of oral and poster presentations and lectures. However, in revised version of the manuscript we extend the section of discussion. It helps to clarify the place of the approach among others as well as to suggest the direction of the MARCS model development.*

*Domínguez, E., and Rivera, H.: A Fokker–Planck–Kolmogorov equation approach for the monthly affluence forecast of Betania hydropower reservoir, J. Hydroinform., 12(4), 486–501, doi: 10.2166/hydro.2010.083, 2010.*

*Shevnina, E.: Deterministic and stochastic models for seasonal forecasting of inflow to reservoirs of hydropower stations, PhD thesis, Russian State Hydrometeorological University, Russia, 188 pp., 2001. (in Russian).*

**Constructive suggestions**
• Author mentioned three statistical moments in line 79. But these are not listed here. For general audience, reviewer suggests to list them.
*Answer: we added the list of the moments in the revised text.*
• Section 1.1 is very rich in mathematical expression. Only audience or practitioner with sound mathematical background easily understands. But for general audience, this section shall be revised in a simpler way…
*Answer: We would like to keep the math "language" of the section, however we arranged the equations on other way: the revised text of the section 1.1 now included only the equations behind the model core, and the Annex provides the theoretical basis for the readers wanted to the details.*
•Please briefly mention what kind of parameters are lumped one and why such is called.
*Answer: we clarify the situation in the revised version of the manuscript.*
• The reviewer wants to have implicit explanation of the secondary parameters like a, b, c and c, Gs. It is not clear how such empirical equations are related with either data or physics.
*Answer: we try to clarify the situation in the revised version of the manuscript.*
• Are the time-series data are daily or monthly or yearly as mentioned in line 227? It would be better to define the time scale.

*Answer: the time series of runoff consists of yearly discharges, thus the time scale of the process considered is multi-year, long term. It was was stressed in the revised version.*

• In order to make the paper strong, reviewer suggests having some key statistics pictorially. This means how the observed set and models are correlated. What is the degree of performance?

*Answer: In this manuscript we presented only the core, not the validation procedure for the probabilistic hydrological model. The validation procedure is described in Shevnina et al. (2017) and includes also figures and tables to show the degree of the model performance under two characterization schemes. Since the text is already long we refrained to add discussion of the model validation block (Shevnina and Gaidukova, 2017).*

•Reviewer feels the paper is somewhat incomplete as in the several statements; the authors did not mention how future works will be proceed. They just envisioned about the future paper.

*Answer: We agree, that steps of the future work were not described in the manuscript, and only the main directions were mentioned. However, now it is still difficult to outline a circle of potential stockholders for a probabilistic form of forecasts of river runoff. This form of forecast allows evaluation of extremes, which is important for risks assessment, in particularly in a design of building construction (Shevnina et al. 2017). In our opinion, it needs to find a common "language" with an Economic, and we have tried to do it in Shevnina et al., 2018. Recently, the direction of the development for the probabilistic hydrological model depends on the Academy of Finland.*

**Specific comments**

• In line 90, comma is needed between features and which.

*Answer: We revised the text.*

• The authors mentioned in parenthesis ("the reference"). What does it mean? It seems the authors forgot to have proper citation. In line 158.

*Answer: We added the explanation.*

• There are three graphs in the paper however, they are not proper captions. In line 273 has Figure 2, but where is Figure 2?

*Answer: We improved the quality of the figure .*

---

## Referee Comment (RC2) · Anonymous Referee #2 · 22 Feb 2019

The article is not original in terms of research methodology. The authors describe the method for assessing the hydrological repercussion of climate change developed at the Department of Hydrophysics and Hydrological Forecasts of the Russian State Hydrometeorological University (RSHU). The theses from the textbook of V.V. Kovalenko are the main content of the article (Modelling of hydrological processes, Gidrometeiz-dat, St. Petersburg, Russia, 1993). It should also be noted that the method from article is taught to students in the undergraduate program RSHU and the method is described in textbooks and methodological recommendations for students with a "step-by-step" algorithm for obtaining results (see for example Practical tasks on the discipline "Hydrological forecasts" http://elib.rshu.ru/files\_books/pdf/rid\_00d41c4c01bd4db
7a25f15faacf9705d.pdf, ÑĄÑĆÑĂ. 24 - 28). The authors assert that this method is presented for the first time in English, contains no typing errors and the calculation formulas are obtained "step by step". These statements can be disproved. 1. There are publications that contain a description of the method under consideration, and in some sources the presentation of method is clearer than in this article. The methodological approach was developed more than 30 years ago. Since that time, it has been tested in many world catchments. Its methodology is applied and developed in countries such as Russia, Colombia, Bolivia, Côte d'Ivoire, Mali, and others. The results are published in journals that are part of the world's scientific bases. The authors probably spent little time to get acquainted with published works. 2. The authors mainly refer to the textbook of V.V. Kovalenko, 1993 (Modelling of hydrological processes, Gidrometeizdat, St. Petersburg, Russia, 1993), but this textbook was complemented and reissued in 2006 (Kovalenko, V. V., Victorova, N. V., Gaydukova, E. V.: Modelling of hydrological processes, the Russian State Hydrometeorological University press, St. Petersburg, Russia, 2006). In the reissued version of the textbook, the typos contained in the 1993 textbook were found and corrected. 3. The algorithm given by the authors skipped some important steps. The main skip is the absence of the dynamic core, from which the stochastic equation is obtained. Probably, the authors have done this intentionally, since it is the dynamic core that causes the discussions.

But the most important remark is that the article proposes to use calculation formulas that can result in unstable solutions, especially about the third moment (skewness coefficient). In 2010, recommendations were issued (Kovalenko, V. V., Victorova, N. V., Gaydukova, E. V., Gromova, M. A., Khaustov, V. A. and Shevnina, E. V.: Guideline to estimate a multi-year runoff regime under non-steady climate to design hydraulic contractions, Russian StateHydrometeorological University Press, St. Petersburg, 2010. http://elib.rshu.ru/files\_books/pdf/img-504161958.pdf) in which a model was presented that allows one to obtain reliable solutions of the Fokker – Planck – Kolmogorov equation. From this model, multiplicative noise was removed, which results in reliable solutions. Also, the stability of solutions can be achieved by transferring the multiplicative
noise to the additive component of the equation or another way – by increasing the number of phase variables taken into account by the model. The authors propose to apply calculation formulas that can give unstable solutions without checking whether it is possible to trust the obtained results. A method for checking the stability of solutions of the this prognostic approach has long been known. I hope that in future studies, the authors will take this into account. In this form, as in the article of the authors, the formulas are dangerous to use because of the probability of obtaining unreliable results.

I believe that the method of scenario assessment of the hydrological consequences of climate change considered in the article is relevant (since the fact of climate change is recognized by the world community and one should be able to assess the consequences of this change), credible if sustainable solutions are obtained (its approbation was carried out on numerous world catchments on retrospective material) and practically important (as it allows to obtain probabilistic characteristics of the hydrological regime).

Please also note the supplement to this comment: https://www.geosci-model-dev-discuss.net/gmd-2018-108/gmd-2018-108-RC2supplement.pdf

---

## Author Comment (AC3) · 1 Mar 2019

Anonymous Referee #2 formulates three issues inherent to the manuscript: it presents the method which is not a new (1), it describes the method well known by the international hydrological community (2) and it does not include discussion on the method's limitations (3). We are following the comments of the Anonymous Referee #2 to present our vision of the tasks of the manuscript.

Anonymous Referee #2: "The article is not original in terms of research methodology. The authors describe the method for assessing the hydrological repercussion of climate change developed at the Department of Hydrophysics and Hydrological ForePrinter-friendly version

casts of the Russian State Hydrometeorological University (RSHU). The theses from the textbook of V.V. Kovalenko are the main content of the article (Modelling of hydrological processes, Gidrometeizdat, St. Petersburg, Russia, 1993). It should also be noted that the method from article is taught to students in the undergraduate program RSHU and the method is described in textbooks and methodological recommendations for students with a "step-by-step" algorithm for obtaining results (see for example Practical tasks on the discipline "Hydrological forecasts" http://elib.rshu.ru/files\_books/pdf/rid\_00d41c4c01bd4db7a25f15faacf9705d.pdf, 24 – 28)."

Authors' comments: The method is not a new (see line 69, p. 3). However, despite the fact that the method has long history in the Russian State Hydrometeorological University, it not yet known in a hydrological modeling community. The method needs to knowledge on the theory of automatic systems, which is not among traditional disciplines for hydrologist and water resources managers. The method still rises many questions from the hydrological modeling community (see the discussions to Shevnina et al., 2017: https://www.hydrol-earth-syst-sci.net/21/2559/2017/hess-21-2559-2017-discussion.html and to Shevnina et al., 2018: https://www.hydrol-earth-syst-sci-discuss.net/hess-2018-473/. The discussion of this "model description paper" is too long because the "an unusual statistical approach" is applied, and the text of the manuscript is "mathematically enriched". In fact, our task is to present the formulas coded in the MARCS model version 0.2, not the AFA method itself. By now, the formulas of the model core version 0.1 is only published in the Annex to Shevnina et al., 2017.

Anonymous Referee #2: "The authors assert that this method is presented for the first time in English, contains no typing errors and the calculation formulas are obtained "step by step". These statements can be disproved. 1. There are publications that contain a description of the method under consideration, and in some sources the presentation of method is clearer than in this article. The methodological approach was developed more than 30 years ago. Since that time, it has been tested in many
world catchments. Its methodology is applied and developed in countries such as Russia, Colombia, Bolivia, Côte d'Ivoire, Mali, and others. The results are published in journals that are part of the world's scientific bases. The authors probably spent little time to get acquainted with published works. 2. The authors mainly refer to the textbook of V.V. Kovalenko, 1993 (Modelling of hydrological processes, Gidrometeizdat, St. Petersburg, Russia, 1993), but this textbook was complemented and reissued in 2006 (Kovalenko, V. V., Victorova, N. V., Gaydukova, E. V.: Modelling of hydrological processes, the Russian State Hydrometeorological University press, St. Petersburg, Russia, 2006). In the reissued version of the textbook, the typos contained in the 1993 textbook were found and corrected. 3. The algorithm given by the authors skipped some important steps. The main skip is the absence of the dynamic core, from which the stochastic equation is obtained. Probably, the authors have done this intentionally, since it is the dynamic core that causes the discussions.

Authors' comments: The main task of the manuscript is to present the formulas for the MARCS model core version 0.2. The FPK approach has much broader framework, and more details will be given in following publications in English. (1). The FPK equation approach is used on hydrological studies of river basins located in Russia, Colombia, Bolivia, Mali, etc. However, the majority of studies are published in Russian only, these studies result to PhD theses defended in the Russian State Hydrometeorological University. We do not include them to the list of references, which is already long. We have refereed to publications in international journals (Viktorova and Gromova, 2008; Domínguez and Rivera, 2010: Kovalenko, 2014: Rosmann and Domínguez, 2017: Shevnina et al., 2017, etc) as well as to original studies in Russian critically needed to formulate the MARCS model core version 0.2 (Kovalenko, 1993; Pugachev et al., 1974). (2). It should be noted, that the reprint of Kovalenko (1993) published in 2006 contains even more typos in the formulas was well as miss meaning statements than the original work. For example, the formulas 4.1 on p. 189 contain the notations for summands that used only once, they are not discussed in the following text. The same formulas are given in p. 245 without these summands. The third equation on p. 247
contains the typo, while this formula is given correctly on p. 191. The statement on p. 191 is that a, b0, b1 and b2 are parameters of the FPK equation, however they are the parameters of the Pearson equation (Andreev et al., 2005). We included some pages from Kovalenko et al. (2006) to see the cases of typo mentioned above a as the supplements to our answers to Anonymous Referee #2). (3). In our opinion, the text contains enough formulas and does not skip critical steps in the narration of the AFA method. In particular, the Eq. 3 and Eq. 4 (p. 4) are dynamic and stochastic equations behind the model MARCS. The notations in the Eq. 3 are differ from the original text in Kovalenko (1993) as well as Kovalenko et al. (2006). It is not clear what Anonymous Referee #2 means while mentioned the "absence of dynamical core"? Is it "core" of the algorithm, the method or the MARCS model? In any case, in the revised version of the Annex. We hope, that it would helps better present the MARCS model core version 0.2.

Anonymous Referee #2: "But the most important remark is that the article proposes to use calculation formulas that can result in unstable solutions, especially about the third moment (skewness coefficient). In 2010, recommendations were issued (Kovalenko, V. V., Victorova, N. V., Gaydukova, E. V., Gromova, M. A., Khaustov, V. A. and Shevnina, E. V.: Guideline to estimate a multi-year runoff regime under non-steady climate to design hydraulic contractions, Russian StateHydrometeorological University Press, St. Petersburg, 2010. http://elib.rshu.ru/files\_books/pdf/img-504161958.pdf) in which a model was presented that allows one to obtain reliable solutions of the Fokker – Planck – Kolmogorov equation. From this model, multiplicative noise was removed, which results in reliable solutions. Also, the stability of solutions can be achieved by transferring the multiplicative noise to the additive component of the equation or another way – by increasing the number of phase variables taken into account by the model. The authors propose to apply calculation formulas that can give unstable solutions without checking whether it is possible to trust the obtained results. A method for checking the stability of solutions of the this prognostic approach has long been known. I hope that

**GMDD**
in future studies, the authors will take this into account. In this form, as in the article of the authors, the formulas are dangerous to use because of the probability of obtaining unreliable results."

Authors' comments: In our opinion, the Anonymous Referee #2 claims to the unstable solutions of FPK equation: infinite of statistical moments of high orders. It limits application of the AFA method (Kovalenko, 1993), and it is discussed in Kovalenko (2004) http://elib.rshu.ru/files\_books/pdf/img-417153826.pdf. The author suggests two ways resulted to the stable solutions of FPK. The first one is introduced in Kovalenko, 2004 and briefly described by Anonymous Referee #2. The second way is given in Kovalenko et al. (2010) and coded in the MARCS model version 0.1 (Shevnina et al., 2017). In the revised version of the manuscript we stressed the limitation of the model core version 0.2 and further direction to the model development.

Anonymous Referee #2: "I believe that the method of scenario assessment of the hydrological consequences of climate change considered in the article is relevant (since the fact of climate change is recognized by the world community and one should be able to assess the consequences of this change), credible if sustainable solutions are obtained (its approbation was carried out on numerous world catchments on retrospective material) and practically important (as it allows to obtain probabilistic characteristics of the hydrological regime)."

Authors' comments: We agree that the AFA method is relevant, however we believe that it needs to better formulations in English to be become well known tool for the international hydrological community.

In summary, three issues inherent to the manuscript should be stressed: it presents the new version (0.2) of the probabilistic MARCS model (1), it describes the formulas of the model version 0.2 together with the limitations inherent current version (2) and possible directions of the MARCS model development (3).

GMDD
Please also note the supplement to this comment: https://www.geosci-model-dev-discuss.net/gmd-2018-108/gmd-2018-108-AC3supplement.pdf

---

## Author Comment (AC4) · 2 Mar 2019

Dear Editor,

We noticed that it was not easy to find Referees for this model description paper. The discussion of the manuscript is started 18 April 2018 and since then many Referee's nominations were rejected. However, the comments of two Anonymous Referees and A. Frolov allow improving the text of the manuscript. The following changes were done in the revised version of the manuscript: 1. We moved the theoretical basis of the AFA to the Annex 1. It allows to introduce the Section 1.1 for general audience in simple way (see the Constructive suggestions by Anonymous Referee #1), and at the same time

to keep the math "language" to present the AFA method, which is now a new (see the comment 1 Anonymous Referee #2). Now, the Annex 1 provides the AFA's theoretical basis for readers wanted for details. 2. We added the Fig. 1 to the section 1.1 to better explain place the model core version 0.2 to the general structure of the probabilistic hydrological model MARCS. The limitations of this version of the model core were explained for general readers (the Discussions section) as well as for practitioners (the Annex 1) according to the comments of two Anonymous Referees and A. Frolov. 3. We extend the section of Discussions to better place the model MARCS among others hydrological models. We have been trying to incorporate as much comments as possible in the revised text of the manuscript. However, some of them were not finally incorporated to keep the structure of the manuscript balanced. In particular, we do not rise the discussion the comment 1 by A.Frolov or suggestions by the Anonymous Referee #2 since it needs to additional references to show regional studies or specific technical papers in Russian. However, four important references in Russian were added to the list of references to discuss the comment 3 of the Anonymous Referee #2.

Our time to improve the manuscript was limited by 10 days since the submission of the comments by the Anonymous Referee #2. Not too much to do essential improvement. We would kindly as to extend of the deadline of the final submission by 13 of March, 2019.

with the best regards Elena Shevnina and Andrey Silaev
* * *

---

## Short Comment (SC2) · 4 Mar 2019

Dear Colleagues, My main remarks on the article by E. Shevnina and A. Silaev are as follows. 1.The equation describing the river runoff must not contain noise c, which is generated by measurement errors. The physical reason for this removal is that measurement errors cannot in any way form the river runoff. Therefore, multiplicative noise should be excluded from equation (4-Shev.-Sil.). 2.The approximate method for solving the FPK equation proposed by V. Kovalenko cannot be considered correct until confirmed by professional mathematicians working in the field of Markov processes. In my humble opinion, it is better to use the proven recommendations contained in the

classic monographs, for example, V.I. Tikhonov and M.A. Mironov "Markov processes" (1977). 3. Paying tribute to the studies obtained by experts from Colombia, Côte d'Ivoire, Mali etc., I dare to draw the attention of E.Shevnina and A.Silaev to my some results published in 2006 and 2011. I used the stochastic differential equation describing river runoff long-term fluctuations in the form $(dq(t))/dt=-kq(t)+k[R(t)-E(t)]$, (1) where $q(t)$ is the river runoff, $k$ is the coefficient in the dependence between the $q(t)$ and the total water reserves $w(t)$ in the catchment area, $q(t) = kw(t)$. The solution for (1) was obtained within the framework of the correlation theory of non-Gaussian random processes. On the basis of this solution, exact analytical dependences between the main statistical characteristics of the river runoff and the corresponding precipitation and evaporation parameters were obtained. Namely, the variance and autocorrelation function, the coefficient of variation, the coefficients of mutual correlation between river runoff and precipitation and the one for the river runoff and evaporation. These formulas can be used to estimate the response of statistical characteristics of runoff to changes in precipitation and evaporation, for example, caused by climate change. Details can be found in the (Frolov, 2006). The discrete modification of model (1) was considered in (Frolov, 2011). I hope that references mention above will help E.Shevnina and A.Silaev to point out the advantages of their model of river runoff in comparison with the results obtained by me about 10 years ago. Respectfully, A.Frolov References Frolov F.V. Dynamic-stochastic modeling of long-term fluctuations in river runoff // Water resources. 2006. Vol.33. âĎŰ5, ŇĂŇĂ. 483-493. Frolov F.V. Discrete dynamic-stochastic model of long-term fluctuations in river runoff // Water Resources. 2011. Vol.38. âĎŰ5, ŇĂŇĂ. 583-592.

Please also note the supplement to this comment:
https://www.geosci-model-dev-discuss.net/gmd-2018-108/gmd-2018-108-SC2-supplement.pdf

---

## Author Comment (AC5) · 13 Mar 2019

**Elena Shevnina and Andrey Silaev**

elena.shevnina@fmi.fi

Received and published: 13 March 2019

A. Frolov has stressed three remarks to the manuscript: (1) the applicability of the Eq. (4) to describe multi-year river runoff, (2) the confirmation for the mathematical formulations, and (3) the limitations of the method and its alternatives.

Remark 1: The equation describing the river runoff must not contain noise c, which is generated by measurement errors. The physical reason for this removal is that measurement errors cannot in any way form the river runoff. Therefore, multiplicative noise should be excluded from equation (4-Shev.-Sil.).

Answer: On our opinion, this remark is not correct, and the "noised" parameter is not generated by the errors inherent to measurements. The parameterization of the model (4) is done with observed time series of river runoff, and the measurements errors do not connected with the model (4).

Remark 2: The approximate method for solving the FPK equation proposed by V. Kovalenko cannot be considered correct until confirmed by professional mathematicians working in the field of Markov processes. In my humble opinion, it is better to use the proven recommendations contained in the classic monographs, for example, V. I. Tikhonov and M. A. Mironov "Markov processes" (1977).

Answer: On our opinion, this remark is not correct since it seems that A. Frolov doubts the professional skills of the authors. In this manuscript, we not only translated the parts of two books by Pugachev et al. (1974) and Kovalenko (1993) but also formulated the material logically to provide the equations for the new core of the hydrological model. The back ground of the authors comes from the Hydrology (the frequency analysis and physical modeling: http://polar-meteorology.fmi.fi/staff/elshe/elshe\_simple.html) and the Radiophysics (the statistical theory of automatic system: https://www.hse.ru/en/org/persons/201924).

Remark 3: Paying tribute to the studies obtained by experts from Colombia, Côted'Ivoire, Mali etc., I dare to draw the attention of E. Shevnina and A. Silaev to my some results published in 2006 and 2011. I used the stochastic differential equation describing river runoff long-term fluctuations in the form (dq(t))/dt=-kq(t)+k[R(t)-E(t)],(1) where q (t) is the river runoff, k is the coefficient in the dependence between the q (t) and the total water reserves w (t) in the catchment area, q (t) = kw (t). The solution for (1) was obtained within the framework of the correlation theory of non-Gaussian random processes. On the basis of this solution, exact analytical dependences between the main statistical characteristics of the river runoff and the corresponding precipitation and evaporation parameters were obtained. Namely, the variance and autocorrelation function, the coefficient of variation, the coefficients of mutual correlation between river

runoff and precipitation and the one for the river runoff and evaporation. These formulas can be used to estimate the response of statistical characteristics of runoff to changes in precipitation and evaporation, for example,caused by climate change. Details can be found in the (Frolov, 2006). The discrete modification of model (1) was considered in (Frolov, 2011). I hope that references mention above will help E. Shevnina and A. Silaev to point out the advantages of their model of river runoff in comparison with the results obtained by me about 10 year sago. Respectfully, A. Frolov References Frolov F. V. Dynamic-stochastic modeling of long-term fluctuations in river runoff // Water resources. 2006. Vol.33. âËĞD ÌŃU5, Ñ ÌĘAÑ ÌĘA.483-493. Frolov F.V. Discrete dynamic-stochastic model of long-term fluctuations in river runoff // Water Resources. 2011. Vol.38. âËĞD ÌŃU5, Ñ ÌĘAÑ ÌĘA. 583-592.

Answer: We have had a look to the method suggested by A. Frolov in the publications 2006 and 2011. We found, that it requires for the time series of evaporation in additional to the time series of river runoff. In the MARCSHYDRO model core version 0.2, the only there non-central moments' estimates are evaluated from the historical observations on runoff. It should be noted, that the observational network on evaporation is less developed compare with hydrological networks, and it rises challenges to apply the method described by A. Frolov. More details on the method presented by A. Frovov can be founded on p. 89 in Kovalenko 2004: http://elib.rshu.ru/files\_books/pdf/img-417153826.pdf together with the discussion on its applicability. We added the detailed answer for the Remark 3 (in Russian) as the Supplement). However, looking toward the model development, we will continue the versioning of the core of the model MARC-SHYDRO and testing them on observations.

Generally, A. Frolov continue the discussion on the limitations of the AFA method, which has a long history including the Supplement (in Russian) to SC1 by 05.08.2018.

Please also note the supplement to this comment: https://www.geosci-model-dev-discuss.net/gmd-2018-108/gmd-2018-108-AC5-

supplement.pdf

---

## Author Comment (AC6) · 13 Mar 2019

Generally, A. Frolov continue the discussion on the limitations of the AFA method, which has a long history: see in the supplement.

Please also note the supplement to this comment:
https://www.geosci-model-dev-discuss.net/gmd-2018-108/gmd-2018-108-AC6-supplement.pdf

---

## Author Comment (AC7) · 13 Mar 2019

**Elena Shevnina and Andrey Silaev**

elena.shevnina@fmi.fi

Received and published: 13 March 2019

We noticed that it was not easy to find Referees for our manuscript. The discussion of the manuscript is started 18 April, 2018 and, since then, many Referee's nominations were rejected. Finally, the comments of two Anonymous Referees and A. Frolov result to the revised text of the manuscript. The following changes were implemented in the new text:

Title: In the new test, the name of the model is the MARCSHYDRO instead of MARCS. We realized that the model MARCS is already exists (http://marcs.astro.uu.se/index.php) with the official name the MARCS – atmospheres.

Then, the abstract was rewrote to stress that our paper introduces a new version of the model. In the revised Introduction, the last paragraph explains our motivations to include the theoretical basis of the model core 0.2 into the model description paper.

Structure: We moved the theoretical basis of the AFA to the Annex 1. On our opinion it allows to introduce the Section 1.1 to "a general audience" (see the Constructive suggestions by Anonymous Referee #1) and, at the same time, to keep the math "language" in description of the AFA method behind the new model MARCSHYDRO model. We supposed, that "general audience" are people working on development and evaluation of numerical models of hydrological system and its components.

Sections' content: To place better the MARCSHYDRO model core version 0.2 into the structure of the model we added the Fig. 1 into the Section 1. Then, we discussed the features of the current version of the MARCSHYDRO model such as the prediction on a climate scale, the low computational costs and the direction toward socio-economic applications in long-term risks assessment. Six blocks of the MARCSHYDRO model were breathy introduced in the revised Section 1. The last paragraph of this section is now discussed the limitations of the current core version of the MARCSHYDRO model according to the comments of two Anonymous Referees and A. Frolov. The specific comments by Anonymous Referee #1 in lines 90, 158, 273 were accounted in the text revision. Section 2 hasn't much changes, it describes details of the model set up, forcing and output for the case of the lijoki River basin.

Discussion: The section of Discussions was extended, and now we have been trying to specify the model ability for an "express analysis" of water extremes in changing climate due to low computational costs and direct connection to social-economical applications. We stressed that the method behind the model is not a new (see the comment 1 the Anonymous Referee #2), but not well known outside the Russian hydrological community due to the lack of publications in international journals. In the revised Discussions we also focused on limitations of the method used as well as on the further development of the MARCSHYDRO model (according the comment 3 of the

Anonymous Referee #1).

Conclusion: We stressed the novelty of the core version 0.2 and gave final remarks about results obtained for the Ilijoki River.

Annexes: Recently, the Annex 1 provides the theory and limitations of the AFA for readers wanted for the details. Two tables with the notations used for the secondary parameters were also added into the Annex 1 according suggestion given by the Anonymous Referee #1

References: We have been trying to incorporate as much references as possible in the revised manuscript. However, some of them were not finally included to the revised manuscript to keep the section of References balanced 50/50 for Russian and English publications. In particular, we do not rise the discussion the comment 1 by A. Frolov (in the RC2) or the suggestions by the Anonymous Referee #2 since it needs to additional references to the regional studies or specific technical papers in Russian. However, three important references in Russian were added to the list of references to discuss the comment 3 of the Anonymous Referee #2.

The following references were added to the list:

1. Veijalainen, N., Korhonen, J., Vehviläinen, B. and Koivusalo, H.L Modelling and statistical analysis of catchment water balance and discharge in Finland in 1951–2099 using transient climate scenarios, Journal of Water and Climate Change, Vol. 3, 55–78, 2012.

2. Willmott, C. J. and Robeson S. M.: Climatologically aided interpolation (CAI) of terrestrial air temperature. International Journal of Climatology, 15(2), 221-229, 1995.

3. Yip, Q. K. Y., Burn, D. H., Seglenieks, F., Pietroniro, A. and Soulis, E. D.: Climate impacts on hydrological variables in the Mackenzie River basin, Canadian Water Resource Journal, 37(3), 209–230, 2012.

4. Kovalenko, V. V.: Partial infinite modelling and forecasting of the process of

river-runoff formation. St. Petersburg, RSHU Publishers, 2004. Available on-line: http://elib.rshu.ru/files\_books/pdf/img-417153826.pdf

5. Sokolovskiy D.L.: River runoff (bases on a theory and methods of calculations). Leningrad, Hydrometeoidat, 540 p. 1968. (in Russian)

6. Shevnina, E.: Long-term assessment of the multi-year statistical characteristics of the maximal runoff under the climate change over the Russian Arctic, doctor of science thesis, Russian State Hydrometeorological University, Russia, 355 pp., 2015. (in Russian).

The following references were excluded from the list:

1. Serinaldi, F. and Kilsby, C. G.: Stationary is undead: Uncertainty dominates the distribution of extremes, Adv. Water Res., 77, 17–36, doi:10.1016/j.advwatres.2014.12.013, 2015.

2. Hamududu, B. and Killingtveit, A.: Assessing of Climate Change Impacts on Global Hydropower, Energies, 5(2), 305–322, doi:10.3390/en5020305, 2012.

3. Obrezkov, V.I. (Eds.): Hydroenergy: a handbook for engineers, Energoizdat, Moscow, 1988. (In Russian).

4. Salvosa, L. R.: Tables of Pearson's Type III Function. Ann. Math. Statist., 1, 191–198, 1930.

5. Shevnina, E.: Changes of maximal flow regime in Arctic, Construction of Unique Buildings and Structures, 7(22), 128–1412014, 2014. (in Russian).

6. Shevnina, E. and Krasikov, A.: The probabilistic hydrological model MARCS (MARkov Chain System): the core code (Version 1.0), doi:10.5281/zenodo.1220096, 2018.

We thank to two Anonymous Referees and A. Frolov for their comments to our manuscript. We hope, that the new text allows better understanding of the MARC-

SHYDRO model specifics as well as our motivations behind the submission this model description paper to GMD. The revised version of the manuscript is included as the Supplement.

with the best regards, Elena Shevnina and Andrey Silaev

Please also note the supplement to this comment: https://www.geosci-model-dev-discuss.net/gmd-2018-108/gmd-2018-108-AC7supplement.pdf

---

## Referee Report (RR1)

Repeated article review
«The probabilistic hydrological model MARCS (MARkov Chain System): the theoretical basis
for the core version 0.2»
Elena Shevnina, Andrey Silaev

The new version of the article includes corrections corresponding to the comments made in the review. Almost all remarks are taken into account. Comments that were not taken into account are very extensive and further large research should be made for them.

The basis of the model considered in the article is the methodology, which has already received wide distribution and, accordingly, approbation on numerous catchments of rivers all over the world. Its reliability is proven and not doubted. There is an officially registered scientific school dealing with issues of dynamic, stochastic and partially infinite modeling of hydrological processes. One of the authors of the article comes from this scientific school.

I would like to comment on other reviews of the article, especially the one in which reviewer shows a negative attitude to the method proposed by the authors.

In hydrology, as in any science, there may be many promising directions. All these directions can be developed, successfully applied in parallel with each other and can compete with each other. But this does not mean at all that one direction may be better reasoned than the other. Simply, they can represent different aspects of the same knowledge. It is impossible to say which is better, which is worse, since they have their own subject areas. It is like a multitude of religions coexisting in society.

Probably reviewer Frolov A. represents the views of a competing scientific school. And in this case, the confrontation in the form of finding errors, inaccuracies, illogical conclusions can last for a very long time.

I believe that the article will be interesting to a wide range of hydrologists, especially representatives of both friendly and competing scientific schools on the subject under consideration. This article should be published, maybe marked as "Discussions".

---

## Author Response (AR2)

Dear Referees,

We are pleasant to receive your comments, questions and suggestions, their allow us to improve the text of the manuscript substantially. We would thank for the interesting discussion. We understand, that we may be not perfect also in the this version of our model description paper, but we hope that the formulas are correctly presented in the text.

with the best regards, Elena Shevnina and Andrey Silaev

[revised manuscript text omitted]
 = \left[ -\left(\bar{c} + \widetilde{c}\left(t\right)\right)Q + \left(\bar{N} + \widetilde{N}\left(t\right)\right) \right] dt , \qquad (A.4)$$

where  $a_0(t) = \overline{c} + \widetilde{c}(t)$  is the stochastic parameter of the system (a "noised" watershed physiography, the inverse of runoff coefficient),  $b_0(t)P = \overline{N} + \widetilde{N}(t)$  is the stochastic input for the system (a "noised" precipitation), and  $a_1 = 1$ . The stochastic components of  $\widetilde{c}(t)$  and  $\widetilde{N}(t)$  are the Gaussian "white noise" with zero means, and their intensities are  $G_{\widetilde{c}} \leftarrow G_{\widetilde{N}}$ . The intensities are mutually correlated as  $K_{\widetilde{c}\widetilde{N}}(\tau) = E(\widetilde{c}(t)\widetilde{N}(t+\tau)) = G_{\widetilde{c}\widetilde{N}}\delta(\tau)$ . It should be noted, that the multiplicative parameter  $\overline{c} + \widetilde{c}(t)$  in Eq. (A.4) is the sum of the constant.  $\overline{c}$  and Gaussian «white noise»  $\widetilde{c}(t)$ , and it may lead to the unstable solutions of the Fokker-Plank-Kolmogorov equation (i.e. in may lead to infinite statistical moments of high orders). It limits the application of the AFA method (Kovalenko, 1993). Kovalenko (2004) suggests two solutions, and we will introduce them in a further paper.

where  $a_0(t) = \overline{c} + \widetilde{c}(t)$  is the stochastic parameter of the system (a "noised" watershed physiography, the inverse of runoff coefficient used by Kovalenko, [1993]),  $b_0(t)P = \overline{N} + \widetilde{N}(t)$  is the stochastic input for the system (a "noised" precipitation), and  $a_i = 1$ . The stochastic components of  $\widetilde{c}(t)$  and  $\widetilde{N}(t)$  are the Gaussian "white noise" with zero means, and their intensities are  $G_{\widetilde{c}} - G_{\widetilde{N}}$ . The intensities are mutually correlated as  $K_{\widetilde{c} \widetilde{N}}(\tau) = E(\widetilde{c}(t)\widetilde{N}(t+\tau)) = G_{\widetilde{c} \widetilde{N}}\delta(\tau)$ . It should be noted, that the multiplicative parameter  $\overline{c} + \widetilde{c}(t)$  in Eq. (A.4) is the sum of the constant  $-\overline{c}$  and Gaussian «white noise»  $\widetilde{c}(t)$ , and it may lead to the unstable solutions of the Fokker-Plank-Kolmogorov equation (i.e. in may lead to infinite statistical moments of high orders). It limits the application of the AFA method (Kovalenko, 1993). Kovalenko (2004)

suggests two solutions, and we will introduce them in a further paper.

**A1.3 The Fokker-Plank-Kolmogorov equation and simplifications**

The Fokker-Plank-Kolmogorov equation can be applied to simulate the probability density function (PDF) for the stochastic Q(t) in Eq. (4) (Kovalenko, 1993; Pugachev, 1974):

380
$$\frac{\partial p(Q,t)}{\partial t} = -\frac{\partial}{\partial Q} (A(Q)p(Q,t)) + 0.5 \frac{\partial^2}{\partial Q^2} (B(Q)p(Q,t)) , \qquad (A.5)$$

where p(Q,t) is the PDF of Q at time t; and the drift coefficient (A(Q)) and diffusion coefficients (B(Q)) are calculated as follows (Kovalenko, 1993; Pugachev, 1974):

$$A(Q) = -\left(\bar{c} - 0.5G_{\widetilde{c}}\right)Q - 0.5G_{\widetilde{c}\,\widetilde{N}} + \bar{N} , \qquad (A.6)$$

$$B(Q) = G_{\widetilde{C}} Q^2 - 2QG_{\widetilde{C}\widetilde{N}} + G_{\widetilde{N}} .$$
(A.7)

385 The analytical solution of Eq. (A.5) is difficult and not always needed for practical applications in water engineering since the PDFs of runoff are modelled from a set of statistical estimators, and the moments are from, among others, van Gelder et

al. (2006). The PDFs are described with the set of moments  $m_k = \int_{-\infty}^{+\infty} Q^k p[Q,t] dQ$  (where k is number of the moment,  $k \to \infty$ ). To obtain the equations for  $m_k$ , both sides of Eq. (A.5) were multiplied by a differentiable function  $\psi(Y)$  and then integrated within limits from  $-\infty$  to  $+\infty$  by Q (however, it is supposed that Q > 0):

$$\frac{d\left(\int_{-\infty}^{+\infty}\psi(Q)p(Q,t)dQ\right)}{dt} = \int_{-\infty}^{+\infty}p(Q,t)A(Q)\frac{\partial\psi(Q)}{\partial Q}dQ + 0.5\int_{-\infty}^{+\infty}p(Q,t)B(Q)\frac{\partial^{2}\psi(Q)}{\partial Q^{2}}dQ$$
(A.8).

Then,  $\psi(Q)$  was replaced with  $\psi(Q) = Q^k$  , and Eq. (A.8) was written as:

$$\frac{dm_k(t)}{dt} = \int_{-\infty}^{+\infty} p(Q,t) A(Q) \frac{\partial(Q^k)}{\partial Q} dQ + 0.5 \int_{-\infty}^{+\infty} p(Q,t) B(Q) \frac{\partial^2(Q^k)}{\partial Q^2} dQ \quad . \tag{A.9}$$

For a stationary random process  $dm_k(t)/dt=0$ , and the drift and diffusion coefficients are constant. Thus, Eq. (A.9) was 395 simplified as follows:

For *k*=1:

$$-\left(\bar{c}-0.5G_{\widetilde{c}}\right)m_1-0.5G_{\widetilde{c}\,\widetilde{N}}+\bar{N}=0.$$
(A.10)

For  $k \ge 2$ :

$$-k(\bar{c}-0.5kG_{\widetilde{c}})m_{k}+k\bar{N}m_{k-1}-k(k-0.5)G_{\widetilde{c}\,\widetilde{N}}m_{k-1}+0.5k(k-1)G_{\widetilde{N}}m_{k-2}=0.$$
(A.11)

400 Further, the summands in Eq. (10–11) were divided by  $(2\overline{c}+G_{\widetilde{c}})$ , and new notations were introduced as suggested in the work of Kovalenko (1993) and Pugachev et al. (1974):

$$a = \frac{G_{\widetilde{c}\,\widetilde{N}} + 2\bar{N}}{2\,\overline{c} + G_{\widetilde{c}}} ; \ b_0 = -\frac{G_{\widetilde{N}}}{2\,\overline{c} + G_{\widetilde{c}}} ; \ b_1 = \frac{2G_{\widetilde{c}\,\widetilde{N}}}{2\,\overline{c} + G_{\widetilde{c}}} ; \ b_2 = -\frac{G_{\widetilde{c}}}{2\,\overline{c} + G_{\widetilde{c}}} .$$

Then, for k = 1, 2, 3, 4 the system of Eq. (A.10–11) includes:

$$m_1(2b_2+1)-a+b_1=0$$
, (A.12)

**405**

415

420

$$3b_2 + 1 m_2 + (2b_1 - a)m_1 + b_0 = 0$$
, (A.13)

$$(4b_2+1)m_3+(3b_1-a)m_2+2b_0m_1=0$$
, (A.14)

$$(5b_2+1)m_4+(4b_1-a)m_3+3b_0m_2=0$$
 (A.15)

The set of four moments (m₁, m₂, m₃, m₄) is sufficient to model distributions from the Pearson equationsystem (Andreev et al., 2005; Elderton and Johnson, 1969). However, in water engineering we usually only use three-parametric probability
distributions fitted to observations (Guidelines, 2004; Guidelines, 1984; Bulletin 17-B, 1982). In this case, Gc << c̄ is assumed, thus it leads to b₂=-Gc/(2c̄+Gc)≈0 and (4b₂+1)≈1, (3b₂+1)≈1, (2b₂+1)≈1. To model the PDFs (or EPCs) of annual runoff within the Pearson Type III distribution, the system of Eq. (A.12–15) is simplified as follows:

$$-a+b_1 = -m_1$$
, (A.16)

$$b_0 + 2m_1 b_1 - am_1 = -m_2 , \qquad (A.17)$$

$$2m_1b_0 + 3m_2b_1 - am_2 = -m_3. (A.18)$$

Denoting  $lk = \begin{pmatrix} -m_1 \\ -m_2 \\ -m_3 \end{pmatrix}$ ,  $x = \begin{pmatrix} b_1 \\ b_0 \\ a \end{pmatrix}$  and  $A = \begin{pmatrix} 1 & 0 & -1 \\ 2m_1 & 1 & -m_1 \\ 3m_2 & 2m_1 & -m_2 \end{pmatrix}$ , the parameters  $a, b_0, b_1$  are calculated as  $x_i = D_i / D$ ,

where *D* is the determinant of matrix *A*, and *Di* is the determinant of the matrix obtained by replacing of the column *i* (1, 2, 3) in matrix *A* by the vector *lk*. Finally, the parameters *a*,  $b_0$ ,  $b_1$  are calculated as follows:

$$b_1 = 0.5 \left( 3m_1m_2 - 2m_1^3 - m_3 \right) / \left( m_2 - m_1^2 \right),$$
 (A.19)

$$b_0 = 0.5 \left( m_1^2 m_2 - 2 m_2^2 + m_1 m_3 \right) / \left( m_2 - m_1^2 \right) , \qquad (A.20)$$

$$a = 0.5 \left( 5m_1m_2 - 4m_1^3 - m_3 \right) / \left( m_2 - m_1^2 \right) .$$
(A.21)

**A1.3 Notations**

There are too many notations used to describe the model's core version 0.2, thus the secondary parameters of equations were grouped by the model behind it. Table A.1 shows the notation and description of the secondary parameters for the linear filter

425 stochastic model. Eq. (A.3) is a simplification of Eq. (A.1) that limits the first order ordinal differential equation. It includes three parameters,  $a_0$ ,  $a_1$  and  $b_0$ , and two of them are assumed to be noised. These noised parameters include a constant component (indicated with a bar) and a Gaussian white noise component (indicated with a tilde) with their own intensities. Table A.1 The notation and description of the parameters for a linear filter stochastic model.

[revised manuscript text omitted]

Notes: Projected mean of air temperature ( $\bar{T}_{pr}$ )\* is needed for a regional parametrization scheme (see details Shevnina,

2011), and these values were not used in the model forcing in the case of the Iijoki River at Raasakka.  $\bar{N}_{pr}$  is the projected mean of annual precipitation amount.

| Value                                             | Reference | Projected period: 2020–2050 |       |       |       |       |       |  |
|---------------------------------------------------|-----------|-----------------------------|-------|-------|-------|-------|-------|--|
|                                                   | period:   | HadGEM2-ES                  |       | Ν     | R     |       |       |  |
|                                                   | 1914–2014 | RCP85                       | RCP45 | RCP26 | RCP85 | RCP45 | RCP26 |  |
| Precipitation, mm year-1                          | 625       | 619                         | 637   | 635   | 737   | 695   | 704   |  |
| Specific discharge, mm year -1         | 380       | 375                         | 386   | 385   | 447   | 421   | 427   |  |
| CV                                                | 0.19      | 0.2                         | 0.19  | 0.19  | 0.16  | 0.17  | 0.17  |  |
| CS                                                | -0.04     | -0.04                       | -0.04 | -0.04 | -0.04 | -0.04 | -0.04 |  |
| Q 10% , m 3 s -1 | 475       | 473                         | 483   | 481   | 527   | 505   | 512   |  |
| Q 90% , m 3 s -1 | 293       | 278                         | 297   | 296   | 331   | 354   | 359   |  |

Table 3. The projected climatology and statistics of annual runoff: a case of the Iijoki River.

Figure 1: The MARCSHYDRO model structure and core versions.